Corrected: Author correction

# An intercross population study reveals genes associated with body size and plumage color in ducks

Zhengkui Zhou [1], Ming Li [2], Hong Cheng[2], Wenlei Fan[1], Zhengrong Yuan[3], Qiang Gao[4], Yaxi Xu[1,2], Zhanbao Guo[1], Yunsheng Zhang[1], Jian Hu[1], Hehe Liu[1], Dapeng Liu[1], Weihuang Chen[2], Zhuqing Zheng[2], Yong Jiang[1], Zhiguo Wen[1], Yongming Liu[5], Hua Chen [5], Ming Xie[1], Qi Zhang[1], Wei Huang[1], Wen Wang [6], Shuisheng Hou[1] & Yu Jiang [2]

Comparative population genomics offers an opportunity to discover the signatures of artificial selection during animal domestication, however, their function cannot be directly revealed. We discover the selection signatures using genome-wide comparisons among 40 mallards, 36 indigenous-breed ducks, and 30 Pekin ducks. Then, the phenotypes are fine-mapped based on resequencing of 1026 ducks from an $F_2$ segregating population generated by wild × domestic crosses. Interestingly, the two key economic traits of Pekin duck are associated with two selective sweeps with fixed mutations. A novel intronic insertion most possibly leads to a splicing change in *MITF* accounted for white duck down feathers. And a putative long-distance regulatory mutation causes continuous expression of the *IGF2BP1* gene after birth which increases body size by 15% and feed efficiency by 6%. This study provides new insights into genotype–phenotype associations in animal research and constitutes a promising resource on economically important genes in fowl.

[1] State Key Laboratory of Animal Nutrition; Key Laboratory of Animal (Poultry) Genetics Breeding and Reproduction, Ministry of Agriculture; Institute of Animal Science, Chinese Academy of Agricultural Sciences, Beijing 100193, China. [2] College of Animal Science and Technology, Northwest A&F University, Yangling 712100, China. [3] College of Biological Sciences and Technology, Beijing Forestry University, Beijing 100083, China. [4] Institute of Genetics and Developmental Biology, Chinese Academy of Sciences, Beijing 100101, China. [5] Beijing Institute of Genomics, Chinese Academy of Sciences, Beijing 100101, China. [6] Research Center for Ecology and Environmental Sciences, Northwestern Polytechnical University, Xi'an 710129, China. These authors contributed equally: Zhengkui Zhou, Ming Li, Hong Cheng, Wenlei Fan, Zhengrong Yuan. Correspondence and requests for materials should be addressed to S.H. (email: houss@263.net) or to Y.J. (email: yu.jiang@nwafu.edu.cn)

Both domestication and subsequent breeding lead to rapid phenotypic evolution driven by artificial selection. Through comparative genomic approaches of wild and domestic populations, a number of selective sweeps have been identified in dog[1], chicken[2], pig[3], rabbit[4], and pigeon[5]. However, because of the lack of genetic mapping of domestic traits, the functions of only a handful of such selected loci have been confirmed. Compared with other domesticated animals, fowls have a short generation interval and excellent reproductive capacity, which enable the production of a large segregating population to facilitate the tracing of phenotypic evolution.

The duck (*Anas platyrhynchos*) was derived from the mallard in 500 BC in central China and is one of the most common domestic fowls[6,7]. In addition to various phenotypically diverse indigenous breeds (Supplementary Table 1), there is also the Pekin duck, the most elite breed, which has undergone intensive artificial selection since the Ming Dynasty (A.D. 1368–1644). As opposed to their wild ancestor, the mallard[8], Pekin ducks show many striking changes such as white plumage which is a favorable feature that meets the demand for white down as a filler for jackets or quilts and makes the carcass easy to clean, extraordinary body size, large deposits of sebum, and excellent egg production performance (Supplementary Table 1). Due to these desirable economic traits, the Pekin duck has become the predominant breed used for meat, feather, and egg production in the global duck industry. Consequently, in addition to economic use, the Pekin duck also provides a powerful system for dissect artificial selection mechanisms in farm animals.

Here, we construct a large mallard × Pekin duck segregating population to facilitate the discovery and characterization of domestication genes. The overlapping regions of selective-sweep mapping and a genome-wide association study (GWAS) not only greatly reduce the false discovery rate of the sweep mapping but also provide an understanding of the potential biological functions of the sweep regions.

## Results

**Genome resequencing and variation.** We anchored the scaffolds on genome assembly BGI_duck_1.0[9] to 31 chromosomes, comprising 29 autosomes and the Z and W chromosomes (Supplementary Table 2), based on the radiation hybrid (RH) map[10]. We also performed whole-genome resequencing of 40 mallards from two independent field collections along their East Asian/ Australian migration route, 36 ducks from 12 indigenous breeds of Southern China, and 30 Pekin ducks from three independent populations at a mean depth of ×10 (Supplementary Fig. 1 and Supplementary Table 3). We identified 12.7 million single nucleotide polymorphisms (SNPs) and 0.83 million insertion/ deletion (indel, ≤6 bp) polymorphisms (Supplementary Table 4 and Supplementary Data 1). Moreover, we generated and sequenced 1026 individuals from F$_2$ segregating population intercrosses between mallards and Pekin ducks (Supplementary Fig. 2), with a mean coverage depth of ×5 to trace and annotate the functional correlates of the selection signatures.

**Population structure.** Principle component analysis (PCA) divided mallards, indigenous breeds, and Pekin ducks into separate clusters (Fig. 1b). There was a significant correlation between the genetic clustering pattern of indigenous duck breeds and the locations where they were collected (Fig. 1a, Supplementary Fig. 1). Additionally, indigenous duck breeds clustered more closely with each other, indicating that they have a closer genetic relationship and probably share a similar domestication history. Although the precise origins of Pekin ducks are unclear[11], our phylogenetic tree (Fig. 1c) suggested that both Pekin ducks

and indigenous breeds originated from a single domestication event. Demographic modeling using ∂a∂i supported the idea that the Pekin duck originated from indigenous breeds of South China (Fig. 1d, Supplementary Figs. 3 and 4, Supplementary Tables 5 and 6, and Supplementary Data 2). In addition, both D-statistics (Supplementary Figs. 5 and 6, Supplementary Table 7) and admixture analysis (*k* = 3, Supplementary Fig. 6) indicated that there was slight gene flow between the Pekin duck and the Gaoyou duck, and the latter are located south of the Grand Canal, which is directly connected to Beijing[11] (Fig. 1a).

**Selective sweeps.** We inferred that the duck has undergone two stages of artificial selection (Fig. 1d and Supplementary Fig. 4): (i) domestication from mallards to indigenous breeds and (ii) improvement from indigenous breeds to Pekin ducks. The whole-genome population differentiation value of the improvement ($F_{ST} = 0.10$) was higher than that of domestication ($F_{ST} = 0.07$), suggesting that there was a bottleneck in the formation of the Pekin duck breed followed by extensive genetic drift or artificial selection during the improvement stage (Fig. 1e and Supplementary Table 8). The finding is consistent with Zeder's view[12] that domesticated fowl likely followed a commensal pathway of domestication.

We scanned the genome for regions with extreme divergence in allele frequency ($F_{ST}$) and the highest differences in genetic diversity (π log ratio) in 40-kb sliding windows in two stages to detect candidate divergent regions (CDRs) on autosomes (Fig. 1e and Supplementary Fig. 7). In total, we identified 123 domestication CDRs (Z test, with $F_{ST} > 0.21$ and π log ratio > 0.70) and 64 improvement CDRs ($F_{ST} > 0.30$ and π log ratio > 1.23) both with a significance level of $P < 0.005$ (Supplementary Fig. 7 and Supplementary Tables 8–10). The values of $F_{ST}$ were significantly higher in the CDRs of the improvement stage than in those of the domestication stage ($P = 3.25 \times 10^{-25}$). Additionally, we found only seven selected regions shared by the domestication and improvement stages (Supplementary Table 11).

We identified 45 domestication CDRs carrying at least five nearly fixed standing variations (derived allele frequency > 0.95) in indigenous breeds (Supplementary Table 9). Two extreme CDRs were found close to the genes in the pathways of "ovarian steroidogenesis" (178 sites, Fig. 1f, Supplementary Table 9) and "neuroactive ligand-receptor interaction" (139 sites, Fig. 1g and Supplementary Table 9), respectively. The result implies that improved reproduction ability and changes in the central nervous system associated with behavioral alterations are common characteristics of animal domestication[1,2,4].

*MITF* **associated with plumage color.** During the improvement stage, we found a CDR located on chromosome 13 that contained 21 fixed mutations ($F_{ST} = 1$) under hard selection (Fig. 2b and Supplementary Table 12), and the Pekin duck population showed the lowest genetic diversity ($π = 2.3 \times 10^{-4}$) in this region (Fig. 2c). This region harbors only one gene, microphthalmia-associated transcription factor (*MITF*) (Fig. 2c), which plays a crucial role in the melanogenesis pathway[13]. Mutations in *MITF* lead to decreased pigmentation and various defects, these defects have been extensively reported in several animal species, such as dogs[14], quails[15], and pigs[16]. To demonstrate the mutational effect of *MITF* on the plumage color in Pekin ducks, we performed a GWAS using 76 colored ducks (40 mallards and 36 indigenous ducks) and 30 white Pekin ducks. Indeed, the *MITF* locus exhibited a strong association peak ($-\log_{10} P = 40.57$), indicating that *MITF* was the only candidate causal gene of the white plumage color of Pekin ducks (Fig. 2a). Furthermore, we found that the inheritance patterns of plumage color conformed to Mendel's

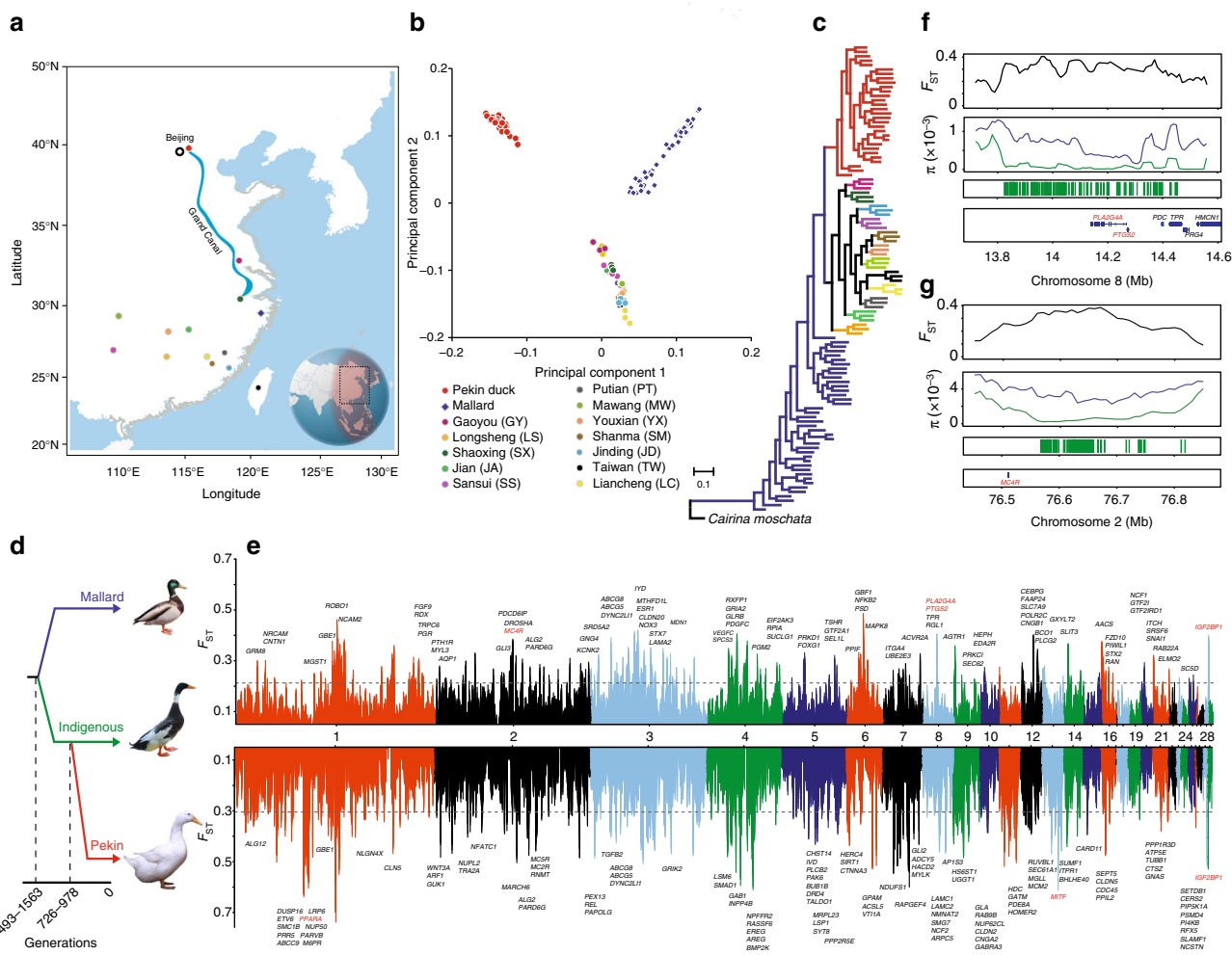

**Fig. 1** Sampling and genomic landscape of the divergence of ducks. **a** The map of duck sampling. Each breed is marked in a different color at its place of origin (Supplementary Table 1). Mallards were sampled in Fenghua City in Zhejiang Province on the East Asian/Australian flyways (indicated by pink shading). The region in the black box represents the region shown in the sampling map. **b** The principal components of the duck samples. **c** Phylogenetic tree (maximum-likelihood (ML) tree with 1000 bootstraps) of all samples inferred from whole-genome tag SNPs (10,240), with Muscovy duck (*Cairina moschata*) as an outgroup. **d** The demographic history of Pekin duck is indicated (Supplementary Figs. 3 and 4), and the generation of divergence is shown at the bottom. **e** Pairwise fixation index ($F_{ST}$) in 40-kb sliding windows across autosomes between mallards and indigenous ducks (top panel; domestication stage) and between indigenous and Pekin ducks (bottom panel; improvement stage). The dashed horizontal line indicates the $F_{ST}$ cutoff (Z test, $P < 0.005$). Genes located in divergence regions annotated by KEGG are indicated by their gene names. **f**, **g** CDRs are enriched for sites on chromosome 8 (**f**, the genes in the pathway of "ovarian steroidogenesis" in red in the bottom panel) and chromosome 2 (**g**, the gene in the pathway of "neuroactive ligand-receptor interaction" in red in the bottom panel) that became nearly fixed (green bars in the third panel) in indigenous breeds in the domestication process

genetic laws in the 1026 wild × domestic $F_2$ generation ducks (colored:colorless = 776:250, $\chi^2_{df = 1} = 0.15$, Fig. 3b and Supplementary Fig. 8). We did not find any fixed SNPs or indels in this region that caused non-synonymous substitutions or frameshift mutations or that were located within conserved sequence elements.

Duck *MITF* consists of two isoforms, *MITF-B* and *MITF-M*, with isoform-specific first exons called 1B and 1M[17]. We identified an ~6.6-kb insertion between exon 1M and exon 2 (Fig. 2d, GenBank: KY114890.1) by comparing the genomes of the Pekin duck and its related species, the goose (*Anser cygnoides*, AnsCyg_PRJNA183603_v1.0)[18]. Then, we analyzed the resequencing data of the 106 ducks and found that this insertion is fixed in Pekin ducks and does not exist in indigenous breeds or mallards (careful examination including additional pooled samples of 60 and 27 mallards from NCBI: SRR3471580-SRR3471606, Supplementary Table 13), which was also

confirmed by PCR (Fig. 2d and Supplementary Fig. 9). This insertion also showed a perfect association with plumage color in the $F_2$ population, as the color of homozygous individuals was completely white and that of heterozygous individuals or individuals without the insertion was colored (Supplementary Figs. 8 and 9). The skin transcriptome data did not show differential expression of *MITF* between Pekin ducks and mallards ($P > 0.05$). However, all the downstream genes of the melanogenesis pathway were significantly down-regulated or even silenced in Pekin ducks (Fig. 2e and Supplementary Data 3), implying that *MITF* is a switch gene.

Large intronic insertions have been reported to change alternative splicing in plants and animals[19,20]. Therefore, the expression differences in three *MITF* exon junctions in the skin of mallards and Pekin ducks were assayed by qPCR (Supplementary Table 14). The results showed that the *MITF-B* isoform was expressed at the same level, whereas Pekin ducks expressed

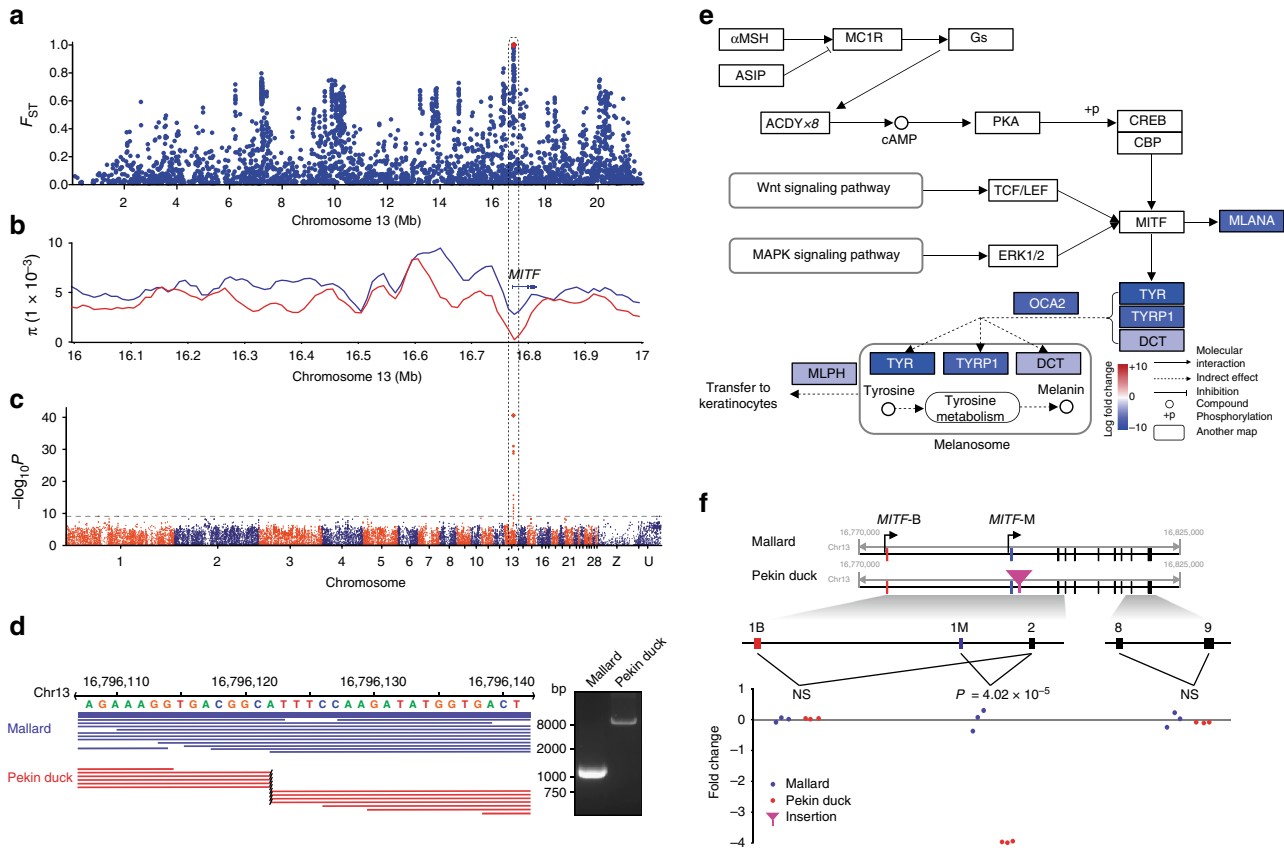

**Fig. 2** Functional genomic basis of the white plumage color of Pekin ducks. **a** Fixation index ($F_{ST}$) of all SNPs along chromosome 13 between mallards and Pekin ducks. Red dots indicate fixed SNPs. **b** The nucleotide diversity ($\pi$) of mallards (blue line) and Pekin ducks (red line) from 16.0 to 17.0 Mb on chromosome 13. **c** GWAS of duck plumage color, including 76 colored ducks and 30 white Pekin ducks. The gray horizontal dashed lines indicate the Bonferroni significance threshold of the GWAS ($1 \times 10^{-9}$). **d** Genomic sequencing read mapping (left) and PCR (right) showing a 6.6-kb intron insertion in Pekin ducks near the 1 M exon of the *MITF* gene. **e** Schematic overview of the melanogenesis pathway, with skin genes showing differential expression between Pekin ducks and mallards. Genes under expressed in mallards are shown in blue (color scale indicates the relative change). **f** Expression differences in three exon junctions between mallards and Pekin ducks according to qPCR. The red triangle represents the intronic insertion on chromosome 13 in Pekin ducks. Exon 1M is specific for the *MITF-M* transcript, while exon 1B is specific for the *MITF-B* transcript. Each exon junction was assayed in three biological replicates with three technical replicates. The indicated P values are based on one-way ANOVA, and NS means nonsignificant

significantly lower levels of the *MITF-M* isoform than mallards ($P = 4.02 \times 10^{-5}$). The *MITF-M* isoform had almost no expression in Pekin ducks (Fig. 2f), suggesting that the splicing changes in *MITF* were most likely caused by a large insertion and resulted in white plumage in Pekin ducks. However, functional verification of this potential causal mechanism is still required. Our study suggested that functional mutations of a critical gene play a prominent role in duck phenotypic evolution[21].

***IGF2BP1* enlarges body size**. A region located at the end of chromosome 28 (4.40–4.71 Mb) (Fig. 3a) showed complete fixation of 18 SNPs ($F_{ST} = 1$) from mallards to Pekin ducks (Supplementary Table 12) and overlapped with both the domestication and improvement CDRs (Fig. 1d). We carried out an additional screen of this region in an independent sample of 87 mallards to confirm that 10 of these SNPs were fixed in Pekin ducks (Supplementary Table 13), but the functional correlates of the sweep remained elusive. We performed a GWAS of all the carcass traits measured in the wild × domestic $F_2$ population. Surprisingly, this sweep overlapped with a quantitative trait locus (QTL) showing a strong association with traits related to body size (body weight, head weight, wing weight, heart weight, liver weight, gizzard weight, leg weight, tarsometatarsus length, and

chest width) (Fig. 3a–c, Supplementary Fig. 10, Supplementary Table 15, and Supplementary Data 4). Linkage analysis showed that 5.39–16.74% of the variance in body size related traits was explained by this genotype, and the effect of a favorable QTL genotype could increase body weight by 15.11% (Fig. 3b and Supplementary Fig. 11 and Supplementary Table 16). Body size is a classic quantitative trait that has received attention from geneticists for more than a century[22]. An extraordinary body size is a critical distinction of Pekin duck that results in better meat performance.

However, candidate genes with high $F_{ST}$ values in this region did not have any non-synonymous or frameshift mutations (Supplementary Table 17). To identify the causal gene, we performed a whole-transcriptome analysis of multiple tissues and different time points between mallards and Pekin ducks (Supplementary Data 3). According to the results of the comparison, only the insulin-like growth factor II mRNA-binding protein 1 (*IGF2BP1*) gene showed significant spatial and temporal differences in expression between mallards and Pekin ducks among the 19 genes located in flanking regions (Fig. 4, Supplementary Fig. 12, and Supplementary Table 18). *IGF2BP1* was consistently expressed in Pekin ducks but was barely expressed in mallards. The expression patterns at 1 day and 2, 4, and 8 weeks after hatching were confirmed by qPCR (Fig. 4a

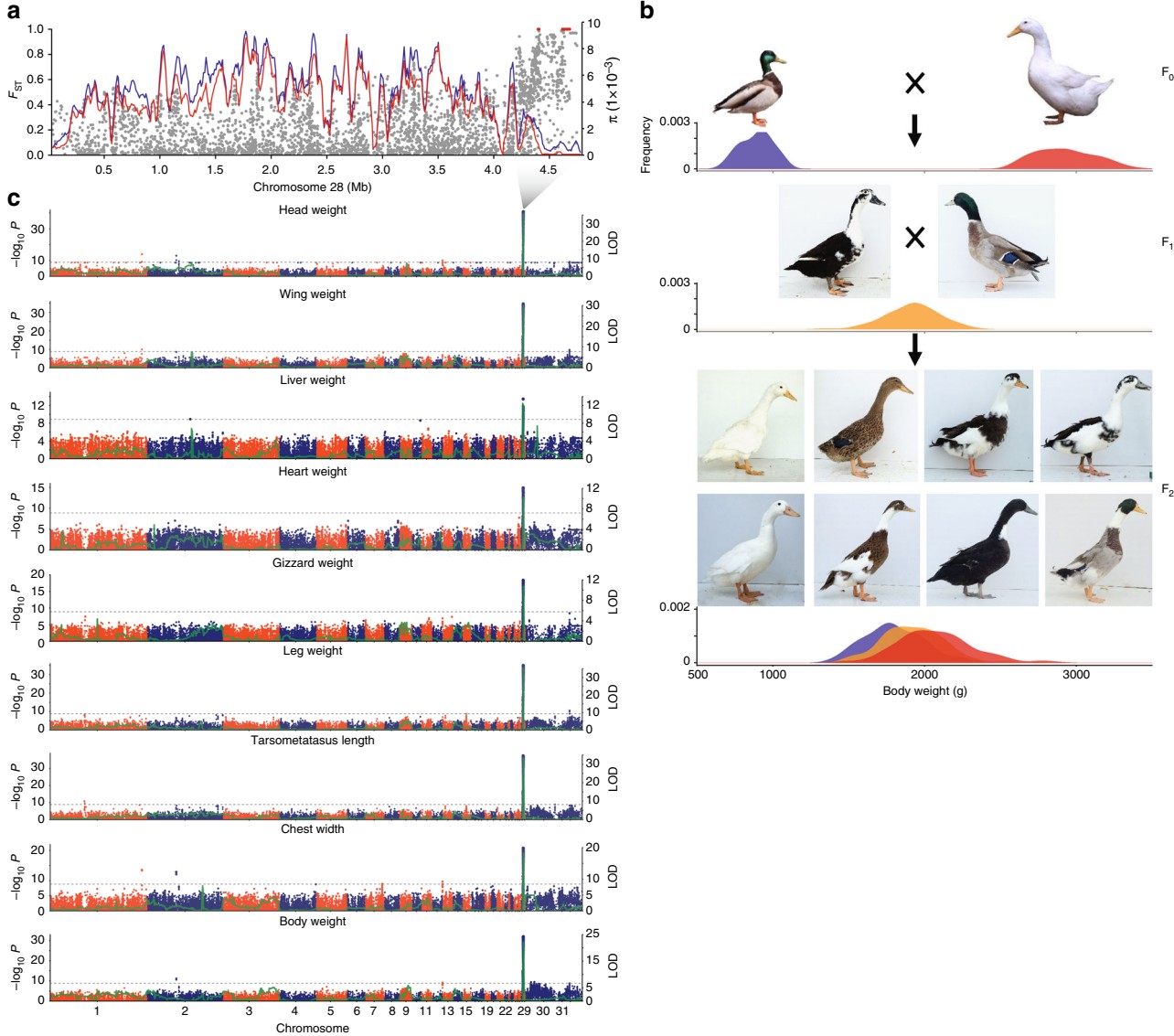

**Fig. 3** Selective sweep associated with body size. **a** Fixation index ($F_{ST}$, gray dots) for all SNPs between mallards and Pekin ducks and nucleotide diversity ($\pi$) in 40-kb sliding windows with 10-kb steps in mallards (blue line) and Pekin ducks (red line) along chromosome 28. Red dots indicate fixed SNPs. **b** Diagram of the $F_2$ population and its body weight and plumage color segregation. The distribution of body weight is presented under the duck images. Blue, red, and orange refer to the mallard, Pekin duck, and heterozygotic haplotypes of *IGF2BP1*, respectively. **c** GWAS and linkage analyses of traits related to duck body size. The green lines refer to the logarithm of the odd (LOD) values. The gray horizontal dashed lines indicate the Bonferroni significance threshold of the GWAS ($1 \times 10^{-9}$)

and Supplementary Tables 14 and 19). Furthermore, immuno-fluorescence histochemistry of liver tissue at 1 day of age showed that Pekin ducks had significantly higher IGF2BP1 protein levels than did mallards (Fig. 4b).

To finely map the causative *cis*-regulatory locus of *IGF2BP1*, we characterized the recombination events in the CDR and identified three recombination breakpoints (Fig. 4c) that divided the 1026 $F_2$ birds into ten haplotypes (Supplementary Figs. 13 and 14). Because the regulatory variation in *IGF2BP1* corresponded to its expression level, we performed qPCR analysis of liver tissue at 8 weeks of age from 121 samples covering the ten haplotypes (Supplementary Tables 14 and 20). We found that the phenotypic values of body size related traits together with *IGF2BP1* expression levels were successfully fine-mapped of the causal variation in an ~100-kb region (chr28: 4,413,785–4,513,671) located on the 148 kb upstream of the *IGF2BP1* gene (Fig. 4c). Considering that errors in scaffold genome assembly, such as

inversions could cause an illusion of long-distance regulation, we examined the end of chromosome 28 using high-throughput chromosome conformation capture (Hi-C) data and collinearity analyses (Supplementary Fig. 15 and Supplementary Table 21), we found a high-quality scaffold order assignment with no evidence of obvious large-scale inversions.

*IGF2BP1* belongs to a family of RNA-binding proteins that are implicated in mRNA localization, turnover, and translational control. *IGF2BP1*-deficient mice show dwarfism, impaired gut development, and downregulation of *IGF2* expression at the embryonic stage[23]. Furthermore, in dogs[24], pigs[25–27], and cattle[28], genes related to the growth hormone/insulin-like growth factor axis have been reported as causal genes of body size changes. This axis is a member of a major growth-promoting signaling system employed by many tissues, and it functions throughout embryonic and postnatal development in an autocrine or paracrine fashion.

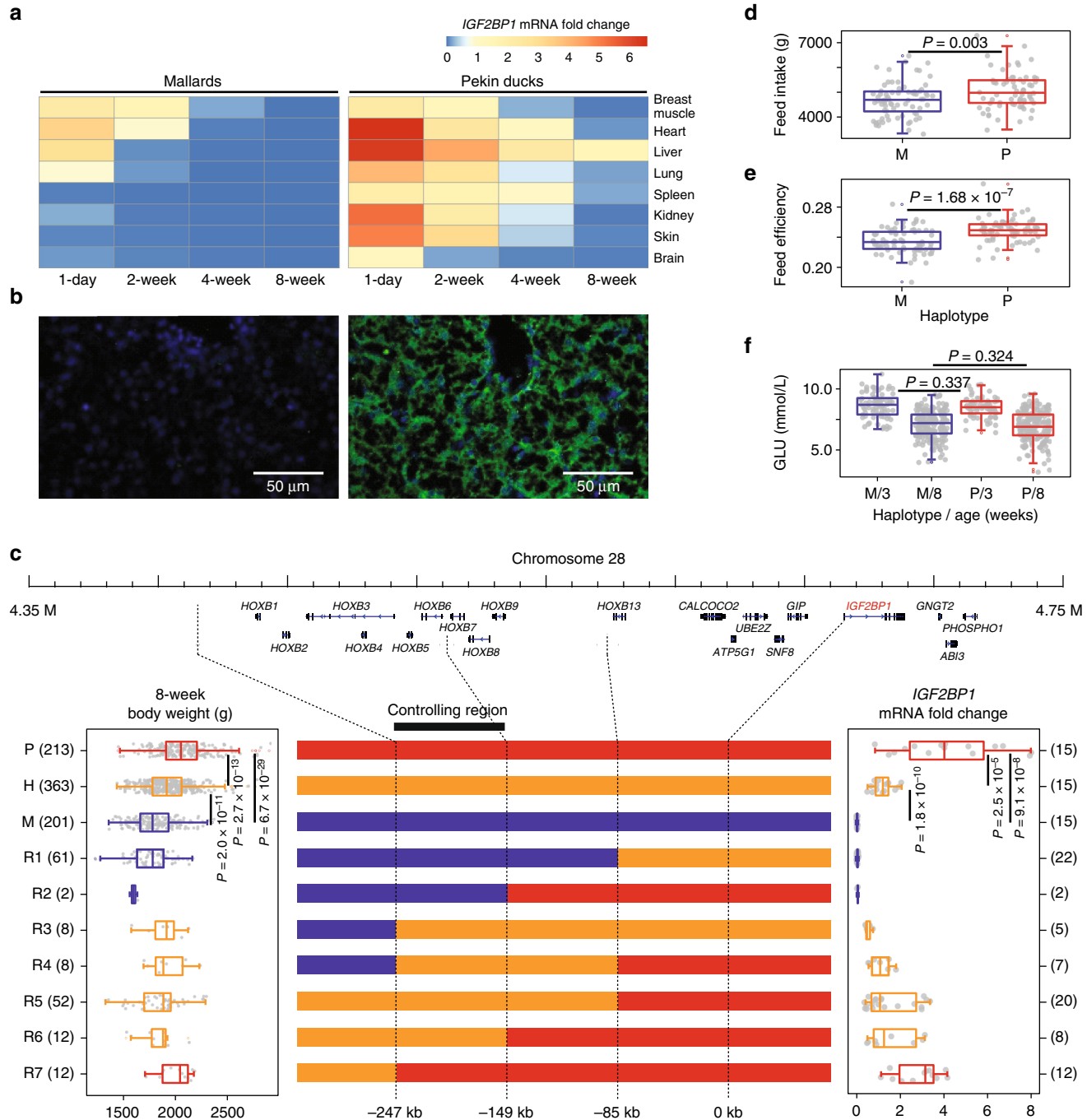

**Fig. 4** Genetic basis of *IGF2BP1* for the enlarged body size of duck. **a** Spatial and temporal expression of *IGF2BP1* in mallards and Pekin ducks measured by qPCR. **b** Immunofluorescence histochemistry staining for IGF2BP1 (green) in liver tissue of mallards and Pekin ducks at 1 day of age. Nuclei were stained with DAPI. **c** Fine mapping of the regulatory region of *IGF2BP1*. Blue, red, and orange bars (center panel) refer to mallard (M), Pekin (P), and heterozygotic (H) genotypes, respectively. R1-7 refer to seven recombinant types. The left and right box plots refer to body weight and *IGF2BP1* expression levels, respectively. The numbers of individuals are given in brackets. The **d** feed intake, **e** feed efficiency, and **f** 3-week and 8-week serum glucose levels were measured in $F_2$ individuals with the mallard (M) and Pekin duck (P) genotypes. The indicated *P* values are based on one-way ANOVA. Box plots denote median (center line), 25–75th percentile (limits), minimum and maximum values without outliers (whiskers), and outliers (red and blue dots)

Finally, we investigated the metabolic mechanism by which *IGF2BP1* changes the body size of ducks. Assuming that there is no population structure and that the feeding conditions are held constant in the segregating population, eating more feed, or/and converting more feed are the main ways to gain weight. Therefore, we measured feed intake and feed efficiency in the wild × domestic $F_2$ population (Fig. 4d, e). Both metrics differed significantly between the individuals with the

ancestral allele and those with the derived allele (one-way ANOVA: feed intake: $P = 0.003$; feed efficiency: $P = 1.68 \times 10^{-7}$). Moreover, the GWAS showed that the *IGF2BP1* locus was strongly associated with feed efficiency ($-\log_{10} P = 12.40$) as well. This locus increased feed intake by 7.5% and feed efficiency by 6% from 3-week to 8-week, suggesting that the effects occurred via multiple downstream pathways regulated by *IGF2BP1*.

Overexpression of *Lin28a* in transgenic mice can up-regulate *IGF2BP1* expression and significantly increase body size. It also reduces serum glucose levels by enhancing glucose uptake in peripheral tissues[29,30]. However, we did not observe a difference in serum glucose levels between individuals with the ancestral and derived alleles in the wild × domestic F$_2$ population (Fig. 4f, one-way ANOVA: 3-week: $P = 0.337$; 8-week: $P = 0.324$), suggesting that different mechanisms may exist in transgenic mice and ducks. Previous studies have shown that *IGF2BP1* is expressed during the embryonic stage and it is essential for early development[23]. Additionally, we observed from public transcriptome data that *IGF2BP1* was not expressed in most postnatal organs of human, mouse, zebrafish, chicken, or mallard, except in some reproductive organs (Supplementary Fig. 16 and Supplementary Table 22). However, the gene was expressed in all organs in Pekin duck after hatching. Hence, our study revealed that a unique mechanism led to the enlarged body size of Pekin ducks, and this finding implies that consistent postnatal expression of *IGF2BP1* in other animals may also enlarge their body size. Therefore, *IGF2BP1* is a strong performance target for meat production and stature improvement in animals.

## Discussion

Our study examined wild, indigenous, and highly selected ducks and presented a complete, systematic, evolutionary route of commensal domestication, and subsequent improvement processes when combined with segregating population analyses. The present results advance our understanding of the genomic basis of critical traits under artificial selection. The selection signatures with fixed variations mainly consisted of standing genetic variations in both stages of the process. However, the novel structural variation in *MITF* led to a breed-specific trait and thus may have contributed to the historical development of the Pekin duck breed. Another putative new mutation near *IGF2BP1*, fixed under continual artificial selection, acted as a major-effect QTL and improved the performance of meat production. These two striking cases strongly highlight our finding that the creation of new traits occurred surprisingly quickly, as it was not believed to be possible in the comparatively short domestication process.

## Methods

**Sampling**. Wild, domestic, and elite duck breeds were collected to identify regions of the genome that were likely to have been targeted during the periods of domestication and improvement. For wild ducks, mallards (*Anas platyrhynchos*) were obtained from the Aoji Duck Farm, Zhejiang Province (Fenghua Institute of Mallards, hunting license available), and we selected 40 individuals (3 male and 37 female ducks) for resequencing. We sampled 12 indigenous breeds that are widely distributed in South China (Supplementary Table 1). Three individuals (including two females) of each indigenous breed were selected for resequencing. Thirty Pekin ducks (15 male and 15 female ducks) were obtained from three independent populations at Beijing Golden Star Duck Co., LTD. A complete description of the appearance, characteristics, and sample distributions of the breeds in this study is presented in Supplementary Table 1 and Supplementary Fig. 1. For all samples, blood was obtained from wing veins and rapidly frozen to −20 °C. Total genomic DNA was then extracted using a traditional phenol-chloroform protocol. We constructed the duck F$_2$ population from 2014 onward at the duck farm of the Institute of Animal Science, Chinese Academy of Agricultural Sciences (Supplementary Fig. 2 and Supplementary Note 1). We performed a slaughter experiment of more than 1000 ducks on the same day and measured a series of traits, including appearance, carcasses, blood biochemical indices, and quality of meat. Furthermore, the blood of all F$_0$, F$_1$, and F$_2$ ducks was obtained for DNA extraction and blood biochemical examinations. Tissues were sampled for RNA or protein extraction and used in transcriptome and proteome analyses. All experiments with ducks were performed under the guidance of ethical regulation from Institute of Animal Science, Chinese Academy of Agricultural Sciences, Beijing, China.

**Chromosome assembly and annotation**. We started the chromosome assembly using the BGI duck 1.0 reference (GCA_000355885.1) based on the RH map[10] with ALLMAP[31]. Scaffolds without gene annotation information or measuring <1 kb were discarded during the process of assembly. We initially assembled chromosome W according to the scaffold order from the bird sex chromosome evolution study[32]. Furthermore, we again adjusted the reference to the copy number variant (CNV) statistics obtained from population resequencing data because half of the read depth of the W chromosome sequence was derived from female individuals. Nonetheless, a number of short scaffolds harboring more than 1000 genes could not be assigned to chromosomes. We merged those unassigned scaffolds as chromosome U (unplaced) for subsequent analysis. Ultimately, a final assembly with a total of 31 chromosomes, including 29 autosomes and chromosomes Z and W, was successfully constructed (Supplementary Table 2). The genome assembly was annotated based on NCBI Annotation Release 100. The coordinate transformation from scaffolds to chromosomes was achieved with LiftOver[33] software. Ultimately, there were 15,912 protein-coding genes annotated in the chromosome-level genome (Supplementary Table 2).

**Whole-genome resequencing**. A total of 106 DNA samples, consisting of 40 mallards, 36 indigenous breeds, and 30 Pekin ducks, were selected from the duck populations described in Supplementary Table 1. Moreover, 1026 individuals were selected from the F$_2$ segregating population. The quality and quantity of DNA were examined using a NanoDrop device and by agarose gel electrophoresis. After the examinations, paired-end libraries were generated for each eligible sample using standard procedures. The average insert size was 500 bp, and the average read length was 150 bp. All libraries were sequenced on an Illumina® HiSeq X Ten or HiSeq 4000 platform to an average raw read sequence coverage of ×10 and ×5 for the natural populations and the 1026 F$_2$ animals, respectively. The depth ensured the accuracy of variant calling and genotyping and met the requirements for population genetic analyses.

**Variant discovery and genotyping**. The 150-bp paired-end raw reads were mapped to the reference genome with Burrows–Wheeler alignment (BWA aln)[34] using default parameters. On average, 85% of the reads were mapped, resulting in a final average sequencing coverage of ×10 (ranging from ×7.5 to ×18.7) per individual. The paired reads that were mapped to the exact same position on the reference genome were removed with MarkDuplicates in Picard[35] to avoid any influence on variant detection. We additionally performed local realignment using GATK[36] to enhance the alignments in regions of indel polymorphisms. After mapping, SNP calling was performed using exclusively GATK[36] (version 3.5), and the output was further filtered using VCFtools[37] (version 0.1.15). SNPs that did not meet the following criteria were excluded: (1) 3 × < mean sequencing depth (over all included individuals) < 30 × ; (2) a minor allele frequency > 0.05 and a max allele frequency < 0.99; (3) maximum missing rate < 0.1; and (4) only two alleles (Supplementary Data 1). Identified SNPs were further classified by SnpEff[38] based on the gene annotation of the reference genome. The process of indel calling was the same as described for SNPs. We strictly filtered the indels in terms of length, and the indels that were ≤6 bp were reserved. The distributions of SNPs and indels in mallards, indigenous-breed ducks, and Pekin ducks are shown in Supplementary Table 3.

**Ancestral state reconstruction**. The ancestral state of the SNPs is important for population and selection analyses[39]. We sequenced a Muscovy duck (*Cairina moschata*) with a sequencing depth of approximately ×10 to obtain an accurate ancestral state of the SNPs in duck. Moreover, we downloaded the resequencing data for 15 goose samples (SRP017498)[40] from NCBI. First, we aligned the short reads of the 16 resequenced individuals with the duck genome using BWA[34] (bwa mem -t 8 -M -R). Next, we used in-house scripts to count the reads that supported the reference allele, alternative allele, and other alleles. In this process, read scores and base scores were required to be greater than 30. Finally, we chose SNPs that had identical states among the 16 individuals as the final ancestral state set (Supplementary Data 2).

**Principle component analysis**. PCA was performed based on all SNPs using EIGENSOFT software (version 4.2)[41,42]. This package applies PCA to genetic data to analyze the population structure. Mallards, indigenous-breed ducks, and Pekin ducks were clearly separated into three groups by the first principal component. The figures were then plotted using the first and second principal components with R packages.

**Phylogenetic analysis**. PLINK[43] was used for pruning pairs in a sliding 50-marker window at 5-marker steps to reduce SNP redundancy caused by linkage disequilibrium (LD) using SNPhylo[44], and 10,240 representative SNPs were extracted. An ML phylogenetic tree was built by the SNPhylo pipeline with standard settings and 1000 bootstrap replicates.

**Structural analysis**. We estimated the ancestry of each individual using the genome-wide unlinked SNP data set and the model-based assignment software program ADMIXTURE 1.3[45] to quantify genome-wide admixture between the mallard, indigenous-breed, and Pekin duck populations. ADMIXTURE was run for each possible group number ($K = 2$ to 6) with 200 bootstrap replicates to estimate the parameter standard errors used to determine the optimal group number ($K$) (Supplementary Fig. 6).

**Demographic history inference using fastsimcoal and ∂a∂i**. We used fastsimcoal2[46] and a diffusion approximation for the demographic inference (∂a∂i)[47] approach to deduce the recent demographic history of duck populations. We chose only SNPs located in intergenic regions to avoid the influence of SNPs under selection, and we also used the folded spectrum method to prevent biases when determining derived allelic states. Comparisons of the allele frequency spectra between model and real data for the three duck populations are presented in Supplementary Fig. 3. A total of 4,466,424 entries in the observed joint site frequency spectrum (SFS) were identified in subsequent simulations. The simulation results of the model are shown in Supplementary Note 2, Supplementary Fig. 4, and Supplementary Table 5.

**ABBA-BABA test (D-statistics)**. We calculated D-statistics[48] to test whether Pekin ducks shared more alleles with mallards or with indigenous breeds using AdmixTools[49]. For the phylogeny (O, ((P1, P2), P3)), there was no gene flow between the outgroup and the other three populations, and the pattern of ABBA therefore reflected gene flow between P1 and P2, whereas the pattern of BABA reflected gene flow between P2 and P3. We calculated the presence of these two patterns as $C_{ABBA}(i)$ and $C_{BABA}(i)$ for site $i$ and then calculated D-statistics using the following equation:

$$D(O, ((P1, P2), P3)) = \frac{\sum_{i=1}^{n}(C_{BABA}(i) - C_{ABBA}(i))}{\sum_{i=1}^{n}(C_{BABA}(i) + C_{ABBA}(i))}$$

A positive value of $D$ is interpreted as gene flow between P1 and P3, whereas a negative value of $D$ is interpreted as gene flow between P2 and P3. In our study, to determine the gene flow between Pekin ducks and indigenous breeds, especially Gaoyou ducks, we used Muscovy ducks (*Cairina moschata*) as the outgroup and Pekin ducks, Gaoyou ducks, and other indigenous breeds as the P1, P2, and P3 groups, respectively. For all comparisons, see Supplementary Table 6.

**Genome scanning for divergent regions**. We detected the CDRs by searching the genome for regions with high fixation index ($F_{ST}$) values[50] and high differences in genetic diversity ($\pi$ ln ratio). First, we calculated the $F_{ST}$ and $\pi$ ln ratio in sliding 40-kb windows with 10-kb steps along the autosomes using VCFtools[37] and in-house scripts for comparisons between mallards and indigenous breeds and between indigenous breeds and Pekin ducks. We then filtered out any windows that had fewer than 40 SNPs in the $F_{ST}$ results. We restricted our CDR descriptions to the windows with a significance level of $P < 0.005$ (Z test) in both $F_{ST}$ and $\pi$ ln ratio, as these windows represented the extreme ends of the distributions (Supplementary Fig. 7, Supplementary Table 7). We avoided the fragmentation of CDRs by performing manual inspections and combining CDRs that were separated by a distance of <200 kb. Finally, we identified 123 and 64 CDRs between mallards and indigenous breeds and between indigenous breeds and Pekin ducks that harbored 533 and 341 genes, respectively. These CDRs composed ~3.0 and 1.5% of the assembled genome for the domestication and improvement stages, respectively (Supplementary Tables 8 and 9; see more at Supplementary Note 3). We scanned the autosomal sites containing standing variations (allele frequency < 0.5) in mallards but nearly fixed (allele frequency > 0.95) variation in indigenous breeds, and identified 2690 sites using this method. Thus, 1407 sites harbored domestication-related CDRs, and 45 of those CDRs contained at least five nearly fixed standing variations (Supplementary Table 8). The top two conspicuous CDRs contained 178 and 139 sites, respectively, and harbored *PLA2G4A* and *PTGS2* in the "ovarian steroidogenesis" pathway (Fig. 1e and Supplementary Tables 8 and 10) and *MC4R* in the "neuroactive ligand-receptor interaction" pathway (Fig. 1f and Supplementary Tables 8 and 10).

**Genome-wide association study**. To minimize false positives and increase statistical power, we considered population structure and cryptic relationships. A mixed linear model program, TASSEL[51], was used for the association analysis. For the $F_2$ population, sex, feeding environment, and forward/backward crosses were set as fixed effects in the mixed model. The kinship derived from whole-genome SNPs of $F_2$ individuals was set as a random effect to control for family effects. We defined the whole-genome significance cutoff as the Bonferroni test threshold and the $F_2$ population threshold as 0.01/total SNPs ($-\log_{10} P = 8.99$).

For the plumage color association in wild mallards, indigenous breeds, and Pekin ducks, the first three PCA values (eigenvectors) derived from whole-genome SNPs represented the fixed effect in the mixed model to correct for stratification[41]. The threshold was 0.01/total SNPs ($-\log_{10} P = 10.10$).

**Linkage analysis**. A large SNP data set from the $F_2$ population was converted to a bin map using a high-density mapping program (in house). The relationship between linkage group and genetic distance was determined by MSTMap software[52], the ML algorithm was used for clustering, and the genetic distance was calculated with the Kosambi model[53]. QTL analysis was performed with the R/qtl package[54], and the composite interval mapping (CIM) analysis method was adopted for QTL mapping. For the LOD threshold screening of 500 permutation tests, the significance level was 0.01, and the significance interval was tested by the Bayesian probability (0.99 confidence level).

**Hi-C experiment and sequencing**. Liver tissues from male Pekin ducks were cross-linked in 20 ml of fresh ice-cold nuclear isolation buffer and 1 ml of ~36% formaldehyde solution under a vacuum for 40 min at room temperature. This reaction was quenched by the addition of 1 ml of 2 M glycine under vacuum filtration for an additional 5 min. The clean samples were ground to powder in liquid nitrogen. The chromatin extraction procedure was similar to that used in the DNase I digestion experiment. The procedures were similar to those described previously[55]. Briefly, chromatin was digested for 16 h with 200 U (4 μl) of *MboI* restriction enzyme at 37 °C. DNA ends were labeled with biotin and incubated at 37 °C for 45 min, and the enzyme was inactivated with 20% SDS solution. DNA ligation was performed by the addition of T4 DNA ligase and incubation at 4 °C for 1 h, followed by 22 °C for 4 h. After ligation, the samples were incubated with proteinase K at 65 °C overnight to reverse the cross-linking. DNA fragments were purified and dissolved in 86 μl of water. Unligated ends were then removed. Purified DNA was fragmented to a size of 300–500 bp, and DNA ends were then repaired. DNA fragments labeled with biotin were finally separated with strepta-vidin C1 beads. Libraries were constructed with an Illumina TruSeq DNA Sample Prep Kit according to the manufacturer's instructions. TA cloning was performed to examine the quality of the Hi-C library. Hi-C libraries were sequenced on an Illumina HiSeq X Ten system. Hi-C was carried out in two independent experiments. The Hi-C experiment and sequencing procedures were performed by Annoroad Gene Technology Co., Ltd., Beijing, China.

**Hi-C data analysis**. Raw Hi-C data were processed to filter out low-quality reads and trim adapters with Trimmomatic[56]. All reads were trimmed to 50 bp, and clean reads were mapped to the duck genome with a two-step approach embedded in the HiC-Pro software[57]. The low-mapping-quality reads, multiple-mapping reads and singletons were discarded. Then, the unique mapping reads were retained in a single file. Read pairs that did not map close to a restriction site or were not within the expected fragment size after shearing were filtered out. Subsequent filtering analyses were performed to discard read pairs from invalid ligation products, including dangling-end and self-ligation products, and from PCR artifacts. The remaining valid read pairs were divided into intrachromosomal pairs and interchromosomal pairs. Contact maps were constructed with chromosome bins of equal sizes for 5 kb, 10 kb, 20 kb, 100 kb, 200 kb, and 500 kb. The raw contact maps were then normalized with a sparse-based implementation of the iterative correction method in HiC-Pro[58] and were visualized by HiTC[59]. Finally, the topologically associated domain (TAD)-like and boundary-like regions were identified with the TopDom method at a resolution of 40 kb[60].

**Structural variation detection**. We found an ~6.6-kb insertion within the *MITF* gene that we proposed to be the causal gene for white plumage in Pekin ducks. We sequenced the PCR product from mallards using Sanger sequencing technology to further correct the gap sequence. Then, we evaluated the genotypes of this structural variation in mallards, Pekin ducks, and $F_2$ individuals from the segregating population using in-house scripts. The input was a BAM file generated by mapping the reads to the corrected genome sequence. The numbers of high-quality reads (−) and malformed reads (+) flanking the structural variation were recorded in the output file for each sample. In addition to the bioinformatics analysis, we further examined the genotypes by PCR. We designed primers using the flanking sequence of the insertion (see Supplementary Table 14 for primer sequence). Considering that ordinary PCR mix is limited in its ability to amplify fragments longer than 2000 bp, KOD FX Neo (TOYOBO, Code No. KFX-201), developed for long fragments, was selected for this experiment. The reaction system was set up and the procedure carried out according to the instructions provided by TOYOBO Company. After 36 cycles, agarose electrophoresis was used to determine the length of the PCR products (Supplementary Fig. 9). We performed the experiments in 100 mallards and 100 Pekin ducks. Furthermore, we performed the same experiment as described above in 50 $F_1$ hybrids and 200 $F_2$ individuals with recorded feather colors. In addition, we retrieved the PCR products from mallards and Pekin ducks and performed Sanger sequencing to confirm whether they were the exact target sequences. The whole sequence of *MITF* in Pekin ducks was submitted to NCBI (KY114890).

**Recombination event analysis of the end of chromosome 28**. We generated an $F_2$ population of 1026 individuals from a cross between mallards and Pekin ducks. We then implemented genotyping at the end of chromosome 28 for the 1026 $F_2$ individuals using the Genome Analysis Toolkit (GATK)[36]. In contrast, according to the previous genotyping results between mallard and Pekin ducks, we obtained 243 SNPs with an absolute allele frequency difference (ΔAF) greater than 0.6 between mallards and Pekin ducks. We identified three recombinant breakpoints across the 243 SNPs, and we then classified the $F_2$ individuals using the three recombinant breakpoints. After a final manual check, we classified the $F_2$ individuals into ten haplotypes (Fig. 3g, Supplementary Fig. 12). In addition, we measured the phenotypes of the $F_2$ individuals (head weight, wing weight, heart weight, liver weight, gizzard weight, leg weight, tarsometatarsus length, chest width, body weight, and feed efficiency). The $F_2$ individuals were fed at three different sites. Moreover, because sex can influence phenotypes, we performed an adjustment procedure using a general linear model (GLM) in R with site and sex as the factors.

In addition, because all $F_2$ individuals used to measure the feed conversion ratio were fed at the same site, sex was the only factor in the adjustment procedure.

**Transcriptome sequencing and analysis**. Multiple tissues (skin, muscle, heart, liver, lung, kidney, spleen, brain, and cartilage) were collected at different ages. Total RNA was isolated with TRIzol reagent (Takara) and then purified for RNA-seq library construction. In total, 41 libraries were finally produced for the RNA-seq experiment and sequenced on an Illumina HiSeq 4000 machine using the 150-bp paired-end sequencing module. The average output was 6 Gb per library. Excluding the 41 libraries produced in this study, we download RNA-seq data for six ducks with abdominal obesity[61] from NCBI. RNA-seq paired-end reads from each of 47 libraries were mapped against the above mentioned Pekin duck reference genome using TopHat[62]. Subsequently, the read counts per million (CPM) values for the genes were obtained by running htseq-count[63], and fragments per kilobase per million values were calculated by TopHat (Supplementary Data 3, Supplementary Notes 4–6).

**Quantitative PCR analysis**. We conducted qPCR mainly for two genes, *MITF* and *IGF2BP1*. Primers were designed with Primer Premier software. The primer sequences and annealing temperatures are shown in Supplementary Table 14. Skin tissues from three Pekin ducks and three mallards were used to assay the expression of three *MITF* exon junctions: (1) exon 1B and exon 2; (2) exon 1 M and exon 2; and (3) exon 8 and exon 9. Complementary DNA synthesis from total RNA and two-step quantitative PCR were performed using the Applied Biosystems QuantStudio system. All samples were assayed in at least three technical replicates. The collected data were analyzed using the $2^{-\Delta\Delta Ct}$ method[64], and all the results were normalized to the duck β-actin gene. For *IGF2BP1*, ducks with different recombination types were selected to measure the expression level of *IGF2BP1*. qPCR was carried out according to the method described above, and the primer sequences and annealing temperatures are shown in Supplementary Table 14. If the recombination types had more than 20 samples, we selected 20 individuals at random to perform the qPCR experiment.

**Immunohistochemistry**. The liver tissues excised from each animal were fixed immediately with 4% paraformaldehyde (PFA) in phosphate-buffered saline (PBS, pH 7.4) for 48 h at room temperature. The samples were dehydrated in an ascending series of 10, 20, and 30% sucrose overnight at 4 °C and subsequently embedded in Tissue-Tek O.C.T. compound (Sakura Finetek, Torrance, CA, USA). Next, the liver tissues were cut serially into 5-μm-thick cross-sections using a cryostat (Leica CM1950, Solms, Germany), air dried, and stored at −20 °C. After being washed in PBS, the frozen tissue sections were subjected to antigen retrieval with 1 N hydrochloric acid (HCl). The sections were then rinsed three times with PBS for 5 min each, blocked with 5% normal goat serum (NGS) with 0.3% Triton X-100 in PBS for 1 h at room temperature, and incubated overnight in a wet chamber at 4 °C with the primary anti-rabbit polyclonal antibody against *IGF2BP1* (1:300, Abcam, Cambridge, UK, catalog #ab82968). The control sections were treated with NGS instead of the specific primary antibody. After being washed three times with PBS, the sections were stained with Alexa Fluor® 488-conjugated goat anti-rabbit immunoglobulin G (IgG) (H+L) secondary antibody (1:200, Abcam, Catalog #ab150077) for 1 h at room temperature, washed another three times with PBS, and then stained with 4′,6′-diamidino-2-phenylindole (DAPI, Sigma-Aldrich, St. Louis, MO, USA) for 10 min at room temperature. After being washed in PBS and mounted with fluorescence anti-fading mounting medium (Sigma-Aldrich, St. Louis, MO, USA), the stained sections were analyzed using a Zeiss laser scanning confocal microscope (LSM710, Carl Zeiss, Jena, Germany) under a ×40 objective.

**Data availability**. All the sequences have been deposited in the Sequence Read Archive (https://www.ncbi.nlm.nih.gov/sra) with the accession codes PRJNA471401 and PRJNA450892. We deposited the genome assembly, all of the sequence data and SNP information in BIG Data Center (http://bigd.big.ac.cn/)[65]. The accession numbers are PRJCA000651, PRJCA000647, and GVM000015. The chromosome-level duck genome, the annotations, and Supplementary Data 1 to 4 are available at www.figshare.com/projects/Duck_Project/24214. All other relevant data are available from the authors.

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

## Acknowledgements

This work was supported by grants from the National Natural Science Foundation of China (31672410 to Z.Z.), the National Scientific Supporting Projects of China (2015BAD03B06 to Z.Z.), the National Thousand Youth Talents Plan (to Y.J.), the China Agriculture Research System of Waterfowl (CARS-42 to S.H. and Z.Z.) and the CAAS-Innovation Team Project(ASTIP-2016-IAS-9, CAAS-XTCX2016010-03). We are grateful to Lizhi Lu (Zhejiang Academy of Agricultural Sciences) and Shihai Dong (Aoji Duck Farm, Zhejiang) for providing the mallard samples, Zhongbin Huang (Waterfowl Storage Center of Shishi in Fujian Province) for the indigenous-breed samples, Shengqiang Hu (Beijing Golden Star Duck Co., LTD.) for the Pekin duck samples, and Guohong Chen and Qi Xu (Yangzhou University) for the Gaoyou duck samples. We thank the members of Jilan Chen, Guiping Zhao, Jie Wen, Benhai Xiong, Xugang Luo, Lupei Zhang, Junya Li, Lin Jiang, Yuehui Ma and Lixian Wang (Institute of Animal Science, CAAS), Dabin Zhang's (China Agricultural University) group for help with the F$_2$ population phenotype data collection. We are grateful to Thomas Faraut, Alain Vignal, and Man Rao (INRA) for sharing the duck RH map with us. We are grateful to Yue Zhao for uploading the sequence data to BIG Data Center. We are grateful to Zhixi Tian, Zheng Wang (Institute of Genetics and Developmental Biology, CAS), Qi Zhou (Zhejiang University), and Lingyang Xu (Institute of Animal Sciences, CAAS) for helpful discussions.

## Author contributions

Z.Zhou. and Yu.J. conceived the project and designed the research. S.H. managed the project. Z.Zhou., M.L., H.Cheng., W.C., and Z.Zheng. performed the bioinformatics analysis. Z.Zhou., S.H., Z.G., W.F., and H.Cheng. constructed the F$_2$ population. Z.Zhou., S.H., W.H., W.F., Z.G., Y.X., Y.Z., J.H., H.L., Yong.J., Z.W., M.X. and Q.Z. collected the F$_2$ population phenotype data. H.Cheng., W.F., and D.L. prepared the DNA samples and performed the laboratory experiment. Z.Y. performed the immunofluorescence histochemistry experiment. Q.G. performed the linkage analysis. Y.L. and H.Chen. performed the phylogenetic analysis. Z.Zhou., M.L., H.Cheng., and Yu.J. wrote the paper. W.W. revised the paper.

## Additional information

**Competing interests:** The authors declare no competing interests.

