## [Peer Review File · Nature Communications]

Reviewers' comments:

Reviewer #1 (Remarks to the Author):

The paper by Zhou et al. describes extensive genomic studies in duck and provide some very interesting findings in relation to genes that have likely been under selection during the formation of the domestic Pekin duck. Unfortunately, the quality of the English language used throughout the manuscript is rather poor and often results in ambiguous description that are not very clear. Furthermore, while the authors present some compelling evidence for selective sweeps based on their Fst analyses, I would have preferred if the authors also had used more formal analyses to identify selective sweep regions using e.g. iHS or EHH methods. I am also not convinced about the Fst thresholds used by the authors (see figure 1d and on page 5, lines 8-9) (Btw: The absence of any page or line numbering doesn't help in providing comments. When in my comments I refer to page numbers, I counted the title page as page number 1). How were these thresholds calculated, and did the authors taken into account multiple testing.

The authors have used a duck RH map to order and anchor the BGI scaffolds to specific chromosomes. The authors should provide a table with order and orientation of all the sequence scaffolds that they ordered and assigned to chromosomes (in Supp table 2, only the size of the chromosomes is provided).

Likewise, the authors describe the identification of 12.7 SNPs. These should be deposited in a public database.

The details about the genome alignments of all sequence reads for mall individuals should be more clearly described (in the supplement). How was alignment of the reads done, how were SNPs identified etc.

Page 4, line 8 " and reveal south China breeds underwent complicated historic migration." I don't see how this conclusion is derived from the results presented by the authors.

Page 4, line 23-24 "more intensive genetic change". I think the results show that there was a strong bottleneck during the development of the Pekin duck. This is not necessarily stronger selection per se, but can be the results of a very small starting population.

The statement on page 7, lines 14-16 " Our study not only" This statement is too general. This might have been the case for some traits and some variants, but most likely the majority, in particular the multifactorial quantitative traits, are based on standing variation. Likewise the statement made in line 23 "10 of these SNPs were new mutations": I fail to see the results that support this. These may well be ancient mutations that existed at a very low MAF.

Page 8, line 2-4 and Supp table 16: No standard deviations are provided.

Page 9, lines 2-6 and Figure 3f: I am not at all convinced by these results. The results only show that this region seems to be a functional domain with multiple interactions, not just between the region identified and the IGF2BP1 gene. Furthermore, although a strong candidate, I don't think the authors unequivocally show that the expression of the IGF2BP1 gene is the causative effect. Furthermore, I doubt that the mapping resolution of an F2 cross is sufficient to pinpoint the effect to a 100 kb region. Finally, the authors should have checked the expression of the IGF2BP1 gene in mallard ducks as well.

Page 10, line 21: "Another new mutation.....". (1) One cannot conclude that this is a NEW mutation and (2) the causal mutation has not unequivocally been identified.

Figure 1: Overlay of PCA and map of China. I think this gives the wrong impression and suggest to remove the map of China.

Figure 3f. I don't think the authors can exclude the other genes (HOX) as the potential causative genes that result in the observed differences.

Below I have listed some examples that illustrate that the English language is unacceptable for publication (in any journal). There are many more, but I list the most obvious ones.

It starts already with the title. I would suggest something like:

Genome analyses of a wild x domestic cross reveals selected genes associated with body size and plumage color of ducks.

Abstract line 4; "segregation" should be "segregating" and the sentence "together with 106 wild and domestic genome comparison" doesn't make sense. Also, the use of "annotated" in line 5 is not appropriate.

Abstract line 3: "Their" seems to refer to "signatures of selection" and these do not have a function ("signatures of selection" describes a process not a functional feature of the genome).

Page 3, line 16: What are "incomparable economic traits"?

Page 4, line 2-3 "1026 F2 segregating population intercrosses" This is not correct the authors used 1026 individual F2 derived from an intercross and not 1026 intercrosses.

Page 4, line 5: delete "has" in "has separated"

Page 5, line 12: "in the domestication stage"

Page 5, line 16: "alternations" meant is probably "alterations" and it should be "improved" not "improve"

Page 5, line 19-20 "close to the genes on the pathway of 'ovarian steroidogenesis'"

Page 5, line 25-26 "was selected in the improvement stage."

Page 6, line 1-2: "may under hard selective sweep that contained"

Page 6, line 3-4: "and Pekin duck population showed the lowest diversity ($n = 2.3 \times 10^{-4}$) in the duck genome"

Page 6, line 12-14: "we confirmed the plumage color trait agrees with Mendelian inheritance in the 1026 wild x domestic F2 generation ducks"

Page 7, line 18 "We found only seven selected regions in the two stages" I assume the authors mean 7 regions that were found in both of the two comparisons made.

Page 7, line 26: "Surprisingly, this sweep is overlapped with the quantitative trait locus (QTLs) that show strong association....." Again multiple errors and this should be corrected to "Surprisingly, this sweep overlaps with a quantitative trait locus (QTL) that shows strong association....."

Page 8, line 11: "casual" should be "causal"

Page 8, line 11-13: "we compared the global transcriptomes data of mallard vs. Pekin duck (Supplementary dataset 3) and identified only insulin-like growth factor II mRNA-binding protein 1 (IGF2BP1) gene that shows significantly differential expression both the spatial and temporal between mallards and Pekin ducks among the 19 genes"

Page 8, line 19: "...have a significant higher" delete "a"

Reviewer #2 (Remarks to the Author):

This is a very interesting study that combines selective sweep analysis on two different duck populations. The way that domestication has proceeded in the duck has allowed a really nice separation of domestication related sweeps from more recent improvement related sweeps. I think this is a very novel addition to the domestication literature, and in fact slightly more could be made of this in the manuscript. The authors also use an F2 intercross to QTL map the traits that have been selected during domestication and use these to verify one of the sweep regions. This represents far greater evidence of function than the usual sweep mapping approach of simply looking at promising annotated genes and pontificating over putative functions that is very limited. The manuscript has some strange paragraph and sentence construction at various places, as well as some odd turns of phrase that could use some additional scrutiny. In general it looks like it has been written in 'Nature' format, but could be better suited changing to a more standard style (intro/ methods/ results/ discussion) that Nature Communications accepts.

Major points

- Far too much is made of the KEGG pathway analysis and the authors use the same rather speculative pontificating that many sweep studies have used when looking at the lists of genes that are selected, namely pull out a lot of GO terms and then cherry pick which ones seem to make a nice story. It is always a very weak line of questioning at best, but here when the authors actually go to the trouble of developing an intercross to test these regions later on in the manuscript, it should be greatly reduced. For instance they select two of 16 and then make the case for behaviour and egg production, but really with such poor annotation in the duck genome (and even in far better annotated genomes this approach is generally specious – see an excellent critique in 'Pavlidis, P. et al. A critical assessment of storytelling: GO categories and the importance of validating genomic scans. *Mol Biol Evol* 29(10):3237-3248, 2012' where they randomly generate selective sweeps in *Drosophila* and then make plausible stories from the KEGG terms they pull out of these random data sets).
- Similar to the above issue, the gene *PPARA* is selected due to a rough hypothesis based on weak annotation arguments. There is no additional evidence for this gene above any of the other sweeps, no overlap apparently with any of the linkage-based QTL analysis. This isn't very persuasive, and should be scaled back or dropped entirely.
- For the *MITF* section, the authors find that they have 21 fixed SNPs in the region, plus a 6kb intronic insertion. They perform some nice experiments with RNA expression to show that although *MITF* is not affected, downstream targets are. I think that this highlights *MITF* as the causal gene rather nicely (especially with the large amounts of sequencing they do on mallards with the colour morph). It is not particularly surprising of course (*MITF* has been demonstrated to affect colour in multiple other species), and they do not really verify that the insertion is indeed the causal mutation. Although further mutation verification experiments (transient transfection, maybe EMSA, etc) seem too onerous to perform given the huge amount of sequencing that has already been performed, they should be more cautious when interpreting causality for the insertion and insert a few more caveats.
- I really like the GWAS (technically an F2 linkage analysis) combined with the sweep regions to actually try and meaningfully understand what the sweep regions potentially control, and I think this should be brought up earlier in the manuscript (certainly in place of the KEGG analysis). Currently just the top hits of the GWAS are listed, but the authors should provide the full details of all QTL identified in these scans for all traits in a table, showing direction of effect, r-squared, additive, dominance etc (there are plots in figure 3c, but these are very small and hard to read). When considering all these QTL found, is there a significant overlap between the QTL detected and the selective sweeps (i.e. are the QTL regions enriched for selective sweeps). This would be very interesting to see, and may also give insights into other sweep regions and their possible effects. Also, no details are given (though maybe I overlooked them somewhere in the supplementary methods, but they appear to be absent) for the actual analysis of this F2 population. If they just analysed it like a typical GWAS this will be erroneous unless they also control for the family effects using a relatedness matrix. This is a linkage rather than a linkage disequilibrium population, so standard QTL map construction and analysis would probably be best in any case.
- The authors look specifically at a 300kb sweep region that overlaps with a QTL for head weight and various other phenotypes, with only 1 of the 19 genes in the region showing differential expression over a range of time points. They use recombinants within the region to narrow down the candidate region controlling *IGF2BP1* down to a 100kb region. This seems persuasive, though they do have rather few of certain recombinant types in several cases (the downstream breakpoint is based on 2 R2 birds and 12 R6 birds). Candidate mutation/ polymorphism still lacking potentially, but this is a very nice example of actually using a sweep region to narrow down a candidate region. Possibly a more in-

depth breakdown of this 100kb region using multiple haplotypes in the different mallard and indigenous bird species could help further (or at least illustrate the region). The figure (figure 3) for this section is very busy and could be done with split up into two – it is very hard to read the recombinants plus their phenotype with the current scale for example.

- The Hi-C analysis is poorly explained and the results are not very clear to interpret. It looks like the entire candidate region is associated in 1 large block. Is this true? If so, how useful is this technique for getting down to candidate SNPs, etc? What else can it tell us? I think the authors should expand on this section a little more, especially as space is not really a problem with the manuscript length as it stands.

- I am not entirely sure if F_{ST} is the best way to detect domestication sweeps, since won't it also be sensitive to changes in the wild population too? The authors currently use F_{ST} and π in combination I believe, how many sweeps are identified if the authors just use π as a basis? This isn't too major an issue as they are currently being more selective by using both criteria of course.

Minor points

- The second paragraph (lines 47-58) is very poorly written, with some very strange sentences and odd construction in general. They also use some odd turns of phrase, which I suggest they remove (line 52 'noble plumage', line 53 'fatten skin', line 54 'incomparable economic traits').

- For the sweep overlaps, the authors fail to mention how many sweeps are shared between the domestication and improvement scans until much later on in the manuscript. This should be mentioned when the sweeps are first discussed.

- The F_{ST} differences (0.10 as compared to 0.07) is taken as evidence for stronger selection during the improvement stage. Would it be possible to test whether this is significant somehow? I.e statistically more sweeps in the improvement stage, etc? As it is, this seems like a rather moderate difference in F_{ST} values. I am also not sure that this result can be taken as 'unintentional lower selection intensity to establish a reciprocal relationship with humans'. In fact the meaning of this sentence is unclear, but it implies that domestication has been centred on behavioural traits, though without any evidence to back it up bar male indigenous birds having green heads and lower production traits, it should really be removed.

- Line 172 'All the record traits' rephrase for clarity.

Reviewer #1 (Remarks to the Author):

Q1: The paper by Zhou et al. describes extensive genomic studies in duck and provide some very interesting findings in relation to genes that have likely been under selection during the formation of the domestic Pekin duck. Unfortunately, the quality of the English language used throughout the manuscript is rather poor and often results in ambiguous description that are not very clear.

Response: Thank you very much for your positive comments and constructive suggestions. We added section headings and subheadings in the results to promote the logical and legible flow of the article. The manuscript has been professionally edited by an English editing service agency, American Journal Experts (AJE).

Q2: Furthermore, while the authors present some compelling evidence for selective sweeps based on their F_{ST} analyses, I would have preferred if the authors also had used more formal analyses to identify selective sweep regions using e.g. iHS or EHH methods. I am also not convinced about the F_{ST} thresholds used by the authors (see figure 1d and on page 5, lines 8-9). How were these thresholds calculated, and did the authors taken into account multiple testing.

Response: We apologize for the unclear description in the methods. The selective sweeps were identified using both F_{ST} and the $\ln \pi$ ratio with the threshold defined by the methods used by Qiu et al. (Nature Communications 2015, 6:10283); these methods were used to calculate the significance of each window with the Z test.

We also performed a permutation test (Huang et al. 2012) to validate the feasibility of the Z test. The details of the permutation test are described as follows: “We further verified whether this threshold was appropriate by performing permutations in F_{ST} analysis. Because the sex chromosomes differ from autosomes in several properties that can affect population genetic estimation (such as effective population size, mutation rate, and recombination rate), we partitioned the genome into autosomes and sex chromosomes to detect outlier windows. First, we shuffled the samples of the two populations and randomly divided them into two new populations.

Then, we calculated windowed F_{ST} values between these two new populations (in sliding 40-kb windows with 10-kb steps) and filtered out the windows with less than 40 SNPs. Afterwards, we recorded the maximum value in all the windows and repeated this process 100 times. Finally, we used the fifth maximum permutation value as the final permutation result (Tang et al. 2016. Supplementary Figure 7 and Supplementary Table 8, see the figure below). The results showed that the threshold defined by the Z test was much greater than the threshold of permutations in each pairwise comparison. This result indicates that using the $P < 0.005$ (Z test) as a threshold can effectively exclude false positives. However, it is possible that windows with lower F_{ST} may also merit further investigation, as such windows may have also contributed to duck domestication.”

Supplementary Figure 7. Genome-wide distribution of the $\pi \ln$ ratio and F_{ST} of 40-kb windows with 10-kb steps across all autosomes. (a) Between mallards and indigenous breeds and (b) between indigenous breeds and Peking ducks. Red dots represent windows fulfilling the selected region requirement (corresponding to a Z test $P < 0.005$). The vertical gray dashed line shows the threshold of the $\pi \ln$ ratio, and the two horizontal gray dashed lines show the threshold of F_{ST} and the permutation threshold with 100 bootstraps.

Supplementary Table 8. The mean and threshold of F_{ST} and the $\pi \ln$ ratio.

Population	Mean Weighted F_{ST}	Mean $\pi \ln$ Ratio	The Threshold of F_{ST} ($P < 0.005$)	The Threshold of $\pi \ln$ ratio ($P < 0.005$)	Fifth Maximum of Permutation Value For F_{ST} ⁴	Top 10 Maximum of Permutation Value For F_{ST}
Mal ¹ vs. Ind ²	0.070332	0.05557588	0.207019	0.695007883	0.158785	0.17619,0.173252,0.166365,0.164028,0.158785,0.157641,0.151817,0.150805,0.138721,0.13601
Ind vs. Pek ³	0.10357	0.190408321	0.30449	1.229149769	0.193225	0.222753,0.220095,0.219102,0.204254,0.193225,0.192792,0.191981,0.190458,0.19019,0.184626

¹Mallard

²Indigenous breeds

³Pekin duck

⁴We performed 100 times permutations and sorted the 100 maximum values from largest to smallest. Finally, we took the fifth value as the final permutation threshold (Tang et al. 2016).

Additionally, following your suggestions, we also used *iHS* and *CLR* methods to scan for selective sweeps. The *iHS* and *CLR* were calculated by *selscan* (Szpiech et al. 2014) and *SweeD* (Pavlidis et al. 2013), respectively. We found 18 and 24 CDRs identified by both F_{ST} and the $\ln \pi$ ratio that could be identified by *iHS* and *CLR* methods, respectively, in the domestication and improvement stage (see figure below). These methods were based on different genetic consumptions or principles, especially *iHS* based on *LD*, which is very sensitive to the recombination rate. Not surprisingly,

the overlap among the CDRs detected by the different statistical approaches was underwhelming. A key showed that only ~ 5.3% loci were identified in three or more large-scale genome-wide scans (Joshua M. Akey. 2009), and Wang presented a similar conclusion in their genome-wide scanning for sweep in chicken (Wang et al. 2016).

We also noticed that iHS and CLR could not identify the $MITF$ locus. But it is not surprised due to both of them have less power to detect a hard sweep, especially for a bottlenecked population (Szpiech et al. 2014; Pavlidis et al. 2013, Jensen. 2014). But it has a classic selective region when we check the SNPs pattern (see figure below). Given that the two regions we examined have notable frequency differences, and F_{ST} is more sensitive to them. We continued to use both F_{ST} and the $\ln \pi$ ratio as the main methods to identify selective sweep in the revised manuscript. Additionally, according to reviewer 2, we reduced the description and KEGG analysis of the results of the selection sweeps, with more focus on the function of the two candidate regions overlapping with the GWAS results.

References:

- [1] Xuehui Huang, et al. A map of rice genome variation reveals the origin of cultivated rice. *Nature*. 2012, 490:497-501
- [2] You Tang, et al. GAPIT Version 2: An Enhanced Integrated Tool for Genomic Association and Prediction. *The Plant Genome*. 2016, doi:

10.3835/plantgenome2015.11.0120.

[3] Zachary A. Szpiech, et al. *selscan: An efficient multithreaded program to perform EHH-based scans for positive selection. Molecular Biology and Evolution. 2014, 31: 2824–2827*

[4] Pavlos Pavlidis, et al. *SweeD: Likelihood-based detection of selective sweeps in thousands of genomes. Molecular Biology and Evolution, 2013, 30:2224–2234*

[5] Joshua M. Akey. *Constructing genomic maps of positive selection in humans: Where do we go from here? Genome Research 2009, 19: 711–722*

[6] Mingshan Wang, et al. *Positive selection rather than relaxation of functional constraint drives the evolution of vision during chicken domestication. Cell Research 2016, 26: 556-573*

[7] Jeffrey D. Jensen. *On the unfounded enthusiasm for soft selective sweeps. Nature Communications. 2014, 5:5281. DOI: 10.1038/ncomms6281*

Q3: The authors have used a duck RH map to order and anchor the BGI scaffolds to specific chromosomes. The authors should provide a table with order and orientation of all the sequence scaffolds that they ordered and assigned to chromosomes (in Supp table 2, only the size of the chromosomes is provided). Likewise, the authors describe the identification of 12.7 SNPs. These should be deposited in a public database.

Response: *Thank you for your suggestion. For the RH map, this work done by Dr. Alain Vignal's group (INRA). They shared the final RH map with us. We have cited their duck RH panel paper (Man Rao, et al. BMC genomics, 2012, 13:513). Because NCBI has phased out support for all non-human organisms in dbSNP and dbVar, we deposited the genome assembly, all of the sequence data and SNP information in another public database, BIG Data Center (<http://bigd.big.ac.cn/>). The accession numbers are PRJCA000651, PRJCA000647 and GVM000015.*

Q4: The details about the genome alignments of all sequence reads for all individuals should be more clearly described (in the supplement). How was alignment of the

reads done, how were SNPs identified etc.

Response: *We apologize that we did not provide the details about the genome alignments of all the sequence reads for all the individuals. The information for mapping rate, sequencing depth and SNPs for all the individuals was merged and presented in Supplementary Table 3. The methods for the read alignment and SNP identification are described in the “Variant discovery and genotyping” methods section.*

Q5: Page 4, line 8 “ and reveal south China breeds underwent uncomplicated historic migration.” I don’t see how this conclusion is derived from the results presented by the authors.

Response: *We apologize for the improper scientific reasoning for the PCA results. We appreciate the reviewer for pointing this out, which provided us a chance to improve the manuscript. We have re-examined the correlation between the geographic distance and genetic distance of indigenous ducks (see Supplementary Figure 1); this figure shows a weak correlation coefficient (Pearson correlation coefficient = 0.185; $P < 1 \times 10^{-11}$). Therefore, we revised this sentence as follows: “Principal component analysis (PCA) separated mallards, indigenous breeds and Pekin ducks into three clusters (Fig. 1b). The correlation is significant between the genetic clustering pattern of indigenous duck breeds and their collected locations (Fig 1a, Supplementary Figure 1). Additionally, indigenous duck breeds clustered more closely with each other, indicating that they have a closer genetic relationship and probably share a similar domestication history.”*

Supplementary Figure 1. Scatterplot of geographic distance and genetic distance. Comparisons within indigenous duck breeds are represented by black dots. The dashed regression line is fitted to the data ($D_{genetic} = 0.0285 + (1 \times 10^{-5}) \times D_{geographic}$, $R^2 = 0.0343$).

Q6: Page 4, line 23-24 “more intensive genetic change”. I think the results show that there was a strong bottleneck during the development of the Pekin duck. This is not necessarily stronger selection per se, but can be the results of a very small starting population.

Response: We agree that the Pekin duck may have undergone a bottleneck. A strong bottleneck effect would cause a dramatic decrease in genomic diversity. However, the average genomic diversity (π) for mallards, indigenous breeds and Pekin ducks is 0.0036, 0.0035, 0.0030, respectively (as shown in the following figure on the left) and indicated no severe change in genomic diversity during the improvement stage. Additionally, the $\partial a \partial i$ results (as shown in the following figure on the right, Supplementary Figure 4) showed that the Pekin duck was not subjected to a very narrow bottleneck. Therefore, we considered that a bottleneck and selection jointly resulted in a greater whole genome population differentiation value of improvement

($F_{ST} = 0.10$) than that of domestication ($F_{ST} = 0.07$).

We revised the part of the main text as follows “...suggesting that there was a bottleneck in the formation of Pekin duck followed by extensive genetic drift or artificial selection during the improvement stage.”

Q7: The statement on page 7, lines 14-16 “Our study not only” This statement is too general. This might have been the case for some traits and some variants, but most likely the majority, in particular the multifactorial quantitative traits, are based on standing variation. Likewise the statement made in line 23 “10 of these SNPs were new mutations”: I fail to see the results that support this. These may well be ancient mutations that existed at a very low MAF. Page 10, line 21: “Another new mutation.....”. (1) One cannot conclude that this is a NEW mutation and (2) the causal mutation has not unequivocally been identified.

Response: We apologize for the ambiguity of the definition. We sequenced a Muscovy duck and two spot-billed ducks and downloaded the resequencing data for fifteen geese (SRP017498) from the National Center for Biotechnology Information (NCBI) website. Using the three sibling species as outgroups, we reconstructed the ancestral genotype of mallards (see “ancestral state reconstruction” methods for detailed

information). The original meaning of “new mutation” in the manuscript referred to the variants that newly originated in a wild mallard population with a very low allelic frequency (<1%) or newly originated in a domesticated duck population. Since the numbers of sequenced mallards is inadequate, we agree with the reviewer that these may be ancient mutations that existed at a very low allele frequency. The word “new” seemed to be too absolute here; however, whether these are new mutations or standing variations will be confirmed in a future study. We would like to use the term **fixed mutations** to replace new mutations at present.

We also revised this statement more specifically as follows: “Our study not only unraveled the leucism mechanism of Pekin duck but also suggested that functional mutations of a critical gene play a prominent role in duck phenotypic evolution.”

Q8: Page 8, line 2-4 and Supp table 16: No standard deviations are provided.

Response: Following your suggestions, we added the standard deviation and coefficient variation of nine body size-related traits in Supplementary Figure 10. Additionally, we plotted the frequency distribution of variation in nine traits of the F₂ population.

Q9: Page 9, lines 2-6 and Figure 3f: I am not at all convinced by these results. The results only show that this region seems to be a functional domain with multiple interactions, not just between the region identified and the IGF2BP1 gene.

Response: We apologize that we over-interpreted the Hi-C results. Due to the limitation of Hi-C resolution (40 kb), we agree with the reviewer that the identified topologically associated domain (TAD)-like regions provide weak evidence of an interaction between IGF2BP1 and their long distance cis regulation region. We also tried to detect the interaction between the two regions using the new method, PSYCHIC (Ron et al. 2017). A total of 85 interactions were detected on Chr 28 (FDR < 0.01), but no interaction was detected around the promoter of IGF2BP1. Given that we just have one timing’s Hi-C data, it may be not enough to identify all the

interactions.

To avoid false long-distance regulation results due to assembly errors, we focused on using Hi-C data to show the genome assembly quality in the revision. We also checked the end of chromosome 28 with emphasis. Additionally, we added collinearity analyses to confirm the results, which indicated that chicken and duck have very conservative collinearity (see Supplementary Figure 15). We further revised the text as follows: “Considering inversion can cause an illusion of long-distance regulation, we checked the end of chromosome 28 using Hi-C data and collinearity analyses (Supplementary Fig. 15) and found a high-quality scaffold order assignment and no obvious inversion.”

Supplementary Figure 15. Chromosome interaction mapping (Hi-C) and collinearity analysis confirm the scaffold assembly of the end of chromosome 28. (a) Whole-genome chromatin interaction. The heat map shows a normalized contact matrix in 40 kb bins, with strong contacts in red and weak contacts in white. (b) Chromatin interaction at the end of chromosome 28. The heat map shows a normalized contact matrix in 10-kb bins. (c) Collinearity between chicken and duck at the end of chromosome 28.

Reference

[1] Gil Ron, et al. Promoter-enhancer interactions identified from Hi-C data using probabilistic models and hierarchical topological domains. *Nature Communications* 2017, 8: 2237.

Q10: Furthermore, although a strong candidate, I don't think the authors unequivocally show that the expression of the IGF2BP1 gene is the causative effect. Figure 3f. I don't think the authors can exclude the other genes (HOX) as the potential causative genes that result in the observed differences.

Response: Thank you for your constructive comment. We agree with the reviewer that the Hi-C analysis cannot exclude HOX genes as potential causative genes. However, our study at least provides four lines of evidence to suggest IGF2BP1 as the strongest candidate gene for directly increasing body size. We emphasize this evidence in the revised manuscript.

First, we were unable to find any nearly fixed non-synonymous variation in the coding region of the 19 genes in the QTL region. Therefore, we checked the expression level of the 19 genes in breast muscle, abdominal adipose, liver, skin, and knee cartilage both in mallards and Pekin ducks, using RNA-sequencing. IGF2BP1 was differentially expressed in abdominal adipose ($P_{FDR}=1.16\times 10^{-19}$), liver ($P_{FDR}=7.19\times 10^{-7}$), skin ($P_{FDR}=0.0026$), and knee cartilage ($P_{FDR}=1.36\times 10^{-9}$) (see Supplementary Figure 12 and Supplementary Table 18). Then, we performed real-time PCR and validated the differential expression of IGF2BP1 in heart, lung, spleen, kidney, and brain as well between mallard and Pekin duck (Figure 4a). We noticed that IGF2BP1 was not differentially expressed in breast muscle, and we found that the IGF2BP1 locus genotype was not associated with breast muscle ($P=4.75\times 10^{-9}$, the whole genome Bonferroni significance threshold is $P<1\times 10^{-9}$) in the F_2 population. Second, the IGF2BP1 expression level in liver was strongly associated with body weight ($P = 9.1\times 10^{-8}$, Figure 4c) and body size-related traits in the F_2 population (Supplementary Figure 14). Third, from the view of known gene function, the dog, pig, and cattle, causal genes (IGF1, IGF2) of body size changes have been reported as those related to the GH/IGF axis, while HOX genes have been proposed to be major players in trunk length diversification in vertebrates, without literature reports of HOX gene regulation of body size. Fourth, a previous study showed that IGF2BP1-deficient mice show a dwarf phenotype during the embryo stage.

Supplementary Figure 12. Gene differential expression between mallard and Pekin duck at the end of chromosome 28 in tissues of (a) breast muscle, (b) abdominal adipose, (c) liver, (d) skin, and (e) knee cartilage. The P value was obtained from mallards vs. Pekin ducks using RNA-seq data (Supplementary Dataset 3). The $-\log(P)$ is shown. P values obtained from the exact test using *edgeR*.

Q11: Furthermore, I doubt that the mapping resolution of an F2 cross is sufficient to pinpoint the effect to a 100 kb region.

Response: For linkage analysis, the means and standard deviation of the recombination event for the F₂ generation individual was 49.23 ± 9.22 (see below figure). We merged all the events from 1026 F₂ samples and identified a total of 3905 bins in the whole genome. We have shown the whole genome bin map for the 1026 F₂ population (see below Supplementary Figure 11). Because the large segregation population accumulated more recombination events, we identified 25 recombination breakpoints on chromosome 28, which greatly facilitated the increased mapping resolution. For body size-related trait mapping, among the 4.35MB ~ 4.75MB region, fortunately, we identified 3 breakpoints. The detailed regional recombination map of 1026 F₂ individuals is shown in Supplementary Figure 13.

Supplementary Figure 11. The bin map of the 1026 F_2 population intercross of mallards and Pekin ducks. The red, orange, and blue refer to Pekin, heterozygosity, and mallard genotypes, respectively. A total of 3,905 bin markers were identified in the whole genome. The width of each column is equal to the genetic distance of each chromosome.

Supplementary Figure 13. Regional recombination map of 1026 F_2 individuals. The top refers to the gene structure of the ends of chromosome 28. The black dashed lines refer to the breakpoints 1, 2, and 3, respectively. The red, orange, blue, and white refer to mallard (M), Pekin (P), heterozygosity (H), and missing genotypes, respectively. R1-7 refer to 7 recombinant types. The abbreviations on the left of the plot refer to different recombinant types. The numbers in brackets refer to the numbers of recombinant

individuals.

Q12: Finally, the authors should have checked the expression of the IGF2BP1 gene in mallard ducks as well.

Response: We apologize for not clearly interpreting the expression of IGF2BP1 in mallards, which was previously shown in the complex Figure 3 and Supplementary Figure 12. Expression detection in mallards both by real-time PCR and RNA-seq has been modified as the new Figure 4a, Supplementary Figure 12, and Supplementary Table 18.

Q13: Figure 1: Overlay of PCA and map of China. I think this gives the wrong impression and suggest to remove the map of China.

Response: We apologize for the confusing figure. We separated Figure 1a containing the map of duck sampling and the principal component plot as Fig 1a and Fig 1b (see below figure).

Below I have listed some examples that illustrate that the English language is unacceptable for publication (in any journal). There are many more, but I list the most

obvious ones.

It starts already with the title. I would suggest something like:

Genome analyses of a wild x domestic cross reveals selected genes associated with body size and plumage color of ducks.

Abstract line 4; “segregation” should be “segregating” and the sentence “together with 106 wild and domestic genome comparison” doesn’t make sense. Also, the use of “annoted” in line 5 is not appropriate.

Abstract line 3: “Their” seems to refer to “signatures of selection” and these do not have a function (“signatures of selection” describes a process not a functional feature of the genome).

Page 3, line 16: What are “incomparable economic traits”?

Page 4, line 2-3 “1026 F2 segregating population intercrosses” This is not correct the authors used 1026 individual F2 derived from an intercross and not 1026 intercrosses.

Page 4, line 5: delete “has” in “has separated”

Page 5, line 12: “in the domestication stage”

Page 5, line 16: “alternations” meant is probably “alterations” and it should be “improved” not “improve”

Page 5, line 19-20 “close to the genes on the pathway of ‘ovarian steroidogenesis’

Page 5, line 25-26 “was selected in the improvement stage.”

Page 6, line 1-2: “may under hard selective sweep that contained”

Page 6, line 3-4: “and Pekin duck population showed the lowest diversity ($\pi = 2.3 \times 10^{-4}$) in the duck genome”

Page 6, line 12-14: “we confirmed the plumage color trait agrees with Mendelian inheritance in the 1026 wild \times domestic F2 generation ducks”

Page 7, line 18 “We found only seven selected regions in the two stages” I assume the authors mean 7 regions that were found in both of the two comparisons made.

Page 7, line 26: “Surprisingly, this sweep is overlapped with the quantitative trait locus (QTLs) that show strong association.....” Again multiple errors and this should be corrected to “Surprisingly, this sweep overlaps with a quantitative trait locus

(QTL) that shows strong association.....”

Page 8, line 11: “casual” should be “causal”

Page 8, line 11-13: “we compared the global transcriptomes data of mallard vs. Pekin duck (Supplementary dataset 3) and identified only insulin-like growth factor II mRNA-binding protein 1 (IGF2BP1) gene that shows significantly differential expression both the spatial and temporal between mallards and Pekin ducks among the 19 genes”

Page 8, line 19: “...have a significant higher” delete “a”

***Response:** We apologize for these mistakes. Thank you very much for your time and patience in reviewing our manuscript. We greatly appreciate it. In the revised manuscript, we tried our best to correct the mistakes, and we also sent the revised manuscript to an English-language editing service, American Journal Experts (AJE), for professional editing.*

Reviewer #2 (Remarks to the Author):

Q1: This is a very interesting study that combines selective sweep analysis on two different duck populations. The way that domestication has proceeded in the duck has allowed a really nice separation of domestication related sweeps from more recent improvement related sweeps. I think this is a very novel addition to the domestication literature, and in fact slightly more could be made of this in the manuscript. The authors also use an F2 intercross to QTL map the traits that have been selected during domestication and use these to verify one of the sweep regions. This represents far greater evidence of function than the usual sweep mapping approach of simply looking at promising annotated genes and pontificating over putative functions that is very limited. The manuscript has some strange paragraph and sentence construction at various places, as well as some odd turns of phrase that could use some additional scrutiny. In general it looks like it has been written in 'Nature' format, but could be better suited changing to a more standard style (intro/ methods/ results/ discussion) that Nature Communications accepts.

Response: Thank you for your positive comments and encouragement. We substantially revised the manuscript based on your suggestions.

Major points

Q2: Far too much is made of the KEGG pathway analysis and the authors use the same rather speculative pontificating that many sweep studies have used when looking at the lists of genes that are selected, namely pull out a lot of GO terms and then cherry pick which ones seem to make a nice story. It is always a very weak line of questioning at best, but here when the authors actually go to the trouble of developing an intercross to test these regions later on in the manuscript, it should be greatly reduced. For instance they select two of 16 and then make the case for behaviour and egg production, but really with such poor annotation in the duck genome (and even in far better annotated genomes this approach is generally specious

– see an excellent critique in ‘Pavlidis, P. et al. A critical assessment of storytelling: GO categories and the importance of validating genomic scans. *Mol Biol Evol* 29(10):3237-3248, 2012’ where they randomly generate selective sweeps in *Drosophila* and then make plausible stories from the KEGG terms they pull out of these random data sets).

- Similar to the above issue, the gene PPARA is selected due to a rough hypothesis based on weak annotation arguments. There is no additional evidence for this gene above any of the other sweeps, no overlap apparently with any of the linkage-based QTL analysis. This isn’t very persuasive, and should be scaled back or dropped entirely.

Response: *We greatly appreciate this reviewer’s suggestions. We eliminated the KEGG analysis and removed the detailed description of the PPARA gene.*

Q3: For the MITF section, the authors find that they have 21 fixed SNPs in the region, plus a 6kb intronic insertion. They perform some nice experiments with RNA expression to show that although MITF is not affected, downstream targets are. I think that this highlights MITF as the causal gene rather nicely (especially with the large amounts of sequencing they do on mallards with the colour morph). It is not particularly surprising of course (MITF has been demonstrated to affect colour in multiple other species), and they do not really verify that the insertion is indeed the causal mutation. Although further mutation verification experiments (transient transfection, maybe EMSA, etc) seem too onerous to perform given the huge amount of sequencing that has already been performed, they should be more cautious when interpreting causality for the insertion and insert a few more caveats.

Response: *Thank you for the comment. We completely agree that the evidence for the hypothesis that the insertion is indeed the causal mutation is not sufficient. According to the reviewer’s suggestions, we revised this part of the main text as follows: “...the MITF-M isoform had almost no expression in Pekin ducks (Fig. 2f), strongly*

*suggesting that the splicing changes in MITF were **most likely** caused by insertion and resulted in white plumage in Pekin ducks.”*

Q4: I really like the GWAS (technically an F2 linkage analysis) combined with the sweep regions to actually try and meaningfully understand what the sweep regions potentially control, and I think this should be brought up earlier in the manuscript (certainly in place of the KEGG analysis).

Response: *Thank you for your suggestion. We replaced the KEGG analysis with selective sweep and GWAS combined analysis. We revised this part of the main text in the introduction as follows: “Here, we constructed a large mallard × Pekin duck segregation population to aid in the discovery and characterization of domestication genes. The overlapped regions of selective-sweep mapping and genome-wide association studies (GWAS) not only greatly reduced the false discovery of sweep mapping but also provided an understanding of the potential biological functions of the sweep regions.”*

Q5: Currently just the top hits of the GWAS are listed, but the authors should provide the full details of all QTL identified in these scans for all traits in a table, showing direction of effect, r-squared, additive, dominance etc (there are plots in figure 3c, but these are very small and hard to read).

Response: *We apologize for not clearly describing the procedure. We updated Supplementary Table 15 and added the additive effect, additive -log P value, dominance effect, dominance -log P value, and marker R^2 for the top ten associated SNPs. In addition, we added the GWAS results for which SNP passed the Bonferroni significance threshold ($P < 1 \times 10^{-9}$) for all traits in Supplementary Dataset 4.*

Q6: When considering all these QTL found, is there a significant overlap between the QTL detected and the selective sweeps (i.e. are the QTL regions enriched for selective sweeps). This would be very interesting to see, and may also give insights into other

sweep regions and their possible effects.

Response: *We agree with the reviewer that it would be interesting to map the QTL on more domestication-related traits such as flying ability or behavior. However, it is very difficult to measure these traits. We tried our best to measure more growth and physiology-related traits. To our surprise, almost all the body size-related traits were associated with the same IGF2BP1 locus and plumage color-associated traits with the MITF locus. Fortunately, these two QTLs perfectly overlapped with the top two selective sweeps.*

Q7: Also, no details are given (though maybe I overlooked them somewhere in the supplementary methods, but they appear to be absent) for the actual analysis of this F2 population. If they just analysed it like a typical GWAS this will be erroneous unless they also control for the family effects using a relatedness matrix. This is a linkage rather than a linkage disequilibrium population, so standard QTL map construction and analysis would probably be best in any case.

Response: *We apologize for the ambiguity regarding this point. We revised this part of the methods for the genome-wide association study as follows: “To minimize false positives and increase statistical power, population structure and cryptic relationships were considered. A mixed linear model program, TASSEL, was used for the association analysis. For the F₂ population, the sex, feeding environment and forward/backward cross were set as fixed effect in the mixed model. The kinship derived from whole-genome SNPs of F₂ individuals was set as a random effect to control the family effects. We defined the whole-genome significance cutoff as the Bonferroni test threshold and the F₂ population threshold as 0.01/total SNPs (-log₁₀ P = 8.99).”*

Q8: The authors look specifically at a 300kb sweep region that overlaps with a QTL for head weight and various other phenotypes, with only 1 of the 19 genes in the region showing differential expression over a range of time points. They use

recombinants within the region to narrow down the candidate region controlling IGF2BP1 down to a 100kb region. This seems persuasive, though they do have rather few of certain recombinant types in several cases (the downstream breakpoint is based on 2 R2 birds and 12 R6 birds). Candidate mutation/ polymorphism still lacking potentially, but this is a very nice example of actually using a sweep region to narrow down a candidate region. Possibly a more in-depth breakdown of this 100kb region using multiple haplotypes in the different mallard and indigenous bird species could help further (or at least illustrate the region). The figure (figure 3) for this section is very busy and could be done with split up into two – it is very hard to read the recombinants plus their phenotype with the current scale for example.

Response: *Thank you for your insightful comments and suggestions. We split Figure 3 into Figure 3 and Figure 4. We also placed the Hi-C figure in the supplementary information (see Supplementary Figure 15) and enlarged the plot of the recombinants and their phenotypes in Figure 4.*

Q9: The Hi-C analysis is poorly explained and the results are not very clear to interpret. It looks like the entire candidate region is associated in 1 large block. Is this true? If so, how useful is this technique for getting down to candidate SNPs, etc? What else can it tell us? I think the authors should expand on this section a little more, especially as space is not really a problem with the manuscript length as it stands.

Response: *We apologize that we over-interpreted the Hi-C results. We agree with the reviewer that the identified topologically associated domain (TAD)-like regions provide weak evidence of an interaction between IGF2BP1 and their long distance cis regulation region. We also tried to detect the interaction between the two regions using the new method, PSYCHIC (Ron et al. 2017). A total of 85 interactions were detected on Chr 28 ($FDR < 0.01$), but no interaction was detected around the promoter of IGF2BP1. Given that we just have one timing's Hi-C data, it may be not enough to identify all the interactions.*

To avoid the false long-distance regulation results due to assembly errors, we

focused on using Hi-C data to show the genome assembly quality in the revision (Supplementary Figure 15). We also emphasized the end of chromosome 28. Furthermore, we added collinearity analyses to validate the results, which indicate that chicken and duck have very conservative collinearity. We revised the text as follows: “Considering inversion can cause an illusion of long-distance regulation, we checked the end of chromosome 28 using Hi-C data and collinearity analyses (Supplementary Fig. 15) and found a high-quality scaffold order assignment and no obvious inversion.”

Q10: • I am not entirely sure if F_{ST} is the best way to detect domestication sweeps, since won't it also be sensitive to changes in the wild population too? The authors currently use F_{ST} and π in combination I believe, how many sweeps are identified if the authors just use π as a basis? This isn't too major an issue as they are currently being more selective by using both criteria of course.

***Response:** Please excuse our poor description of this point. We used both F_{ST} and the $\ln \pi$ ratio in our previous manuscript to identify the selective sweeps. For more details, please see our above response.*

Minor points

Q11: The second paragraph (lines 47-58) is very poorly written, with some very strange sentences and odd construction in general. They also use some odd turns of phrase, which I suggest they remove (line 52 ‘noble plumage’, line 53 ‘fatten skin’, line 54 ‘incomparable economic traits’).

***Response:** Thank you for your advice. We corrected the odd turns of phrases and elusive sentences in the revised version. We believe the writing has been greatly improved through the revising process.*

Q8: For the sweep overlaps, the authors fail to mention how many sweeps are shared

between the domestication and improvement scans until much later on in the manuscript. This should be mentioned when the sweeps are first discussed.

Response: Thank you for your suggestion. We changed the order in the revision.

Q12: The F_{ST} differences (0.10 as compared to 0.07) is taken as evidence for stronger selection during the improvement stage. Would it be possible to test whether this is significant somehow? I.e statistically more sweeps in the improvement stage, etc? As it is, this seems like a rather moderate difference in F_{ST} values. I am also not sure that this result can be taken as ‘unintentional lower selection intensity to establish a reciprocal relationship with humans’. In fact the meaning of this sentence is unclear, but it implies that domestication has been centered on behavioral traits, though without any evidence to back it up bar male indigenous birds having green heads and lower production traits, it should really be removed.

Response: We apologize for the ambiguity of the sentences. We revised the first sentence as follows: “...suggesting that there was a bottleneck in the formation of Pekin duck followed by extensive genetics drift or artificial selection during the improvement stage.” We calculated the significance of F_{ST} in the CDRs between the domestication and improvement stage, and the results were statistically significant (see the figure below).

We also revised the text as follows: “it implies that domestication has been centered on behavioral traits” and deleted the following description in the revision.

Q13: Line 172 'All the record traits' rephrase for clarity.

Response: *Thank you for your suggestion. All the recorded traits refer to the carcass traits measured in the F2 population slaughter experiment, such as body weight, head weight, wing weight, heart weight, liver weight, gizzard weight, leg weight, tarsometatarsus length, and chest width. This was clarified in the main text.*

List of the updated figures

Current Version	Last Version
Figure 1	Figure 1
Figure 2	Figure 2
Figure 3	Figure 3
Figure 4	Figure 3
Supplementary Figure 1	Newly added
Supplementary Figure 2	Supplementary Figure 2
Supplementary Figure 3	Supplementary Figure 3
Supplementary Figure 4	Supplementary Figure 4
Supplementary Figure 5	Supplementary Figure 5
Supplementary Figure 6	Supplementary Figure 6
Supplementary Figure 7	Supplementary Figure 7
Supplementary Figure 8	Supplementary Figure 8
Supplementary Figure 9	Supplementary Figure 9
Supplementary Figure 10	Newly added
Supplementary Figure 11	Supplementary Figure 10
Supplementary Figure 12	Supplementary Figure 11
Supplementary Figure 13	Supplementary Figure 12
Supplementary Figure 14	Supplementary Figure 13
Supplementary Figure 15	Newly added
Supplementary Figure 16	Supplementary Figure 14

Reviewers' comments:

Reviewer #1 (Remarks to the Author):

The authors have very much improved the manuscript and addressed most of my comments. The paper now reads very well and I very much like the extensive results presented by the authors. Nevertheless, I still am not convinced by the conclusion from the authors that they have conclusive evidence that a mutation in a regulatory region affecting the IGF2BP1 gene is the causative mutation for the phenotypic differences on growth identified between Peking duck and mallard. On page 8, line 25 they refer to the Suppl. Table 14. However this table only shows the primers used for the qPCR but NOT the expression levels for the 10 different haplotypes. I could not find any information about the actual gene expression within the documents provided. Furthermore, the 100 kb region contains many other genes (HOX genes) and the authors do not show a direct correlation between the phenotype and the gene expression of IGF2BP1. In their cross, there is much LD and they cannot rule out that other genes underlie their observed phenotypic difference. In my view the authors need to (1) provide the gene expression data for the 10 haplotypes and (2) tone down their claims that a mutation in a regulatory sequence for the IGF2BP1 is the causative mutation. In the discussion (page 10, line 20) the authors refer this as "Another putative new mutation in the long-distance regulatory region". Assuming that this region is indeed a long-distance regulator (of which I am not convinced), then they still do not identify the actual mutation within the 100 kb of sequence.

The English language has now been very much improved, but I have identified a few minor mistakes which are listed below.

Page 2, line 5: "segregated for fine mapping". To me this still doesn't sound correct

Page 3, line 6: "fowl" should be "fowls"

Page 3, last sentence: Z and W chromosomes (plural)

Page 4, line 4: replace "for a mean depth" by "at a mean depth". I also think that the word "coverage" is redundant and should be removed.

Page 6, line 9: "association" instead of "associate"

Page 7, line 5: replace "was" by "is"

Page 7, line 23: change to: "the carcass traits measured in the"

Page 10, line 21-22: "which is fixedproductive performance". This still is a weird sentence: "fixed under continual selection as a major effect QTL"? and "reinvented production performance"?

Page 11, line 15: replace "shared" by "for sharing"

Reviewer #2 (Remarks to the Author):

I have now looked over the revisions and most have been met to my satisfaction. A few rather minor points are indicated below, however.

Q3. For the sentence that has been revised "the MITF-M isoform has almost no expression in ducks, strongly suggesting that the splicing changes in MITF were most likely caused by insertion and resulted in white plumage in Pekin ducks", reword to the following:

"the MITF-M isoform has almost no expression in ducks, suggesting that the splicing changes in MITF were most likely caused by insertion and resulted in white plumage in Pekin ducks. However functional verification of this potentially causal mechanism is still required."

Q5. I couldn't supplementary dataset 4 for the full GWAS results, but in any case, these should be

given in a table in the supplementary section.

Q10. I don't think the authors properly understood my question. As I understand it, they classed sweeps as having to meet both FST and pi thresholds. However, how many sweeps would be detected if they just used pi (i.e. where a lot of sweeps discarded as they met the pi requirements but not the FST thresholds)?

Q12. The authors show that the difference in FSTs between domestication and improvement stages is significant, but this p-value should be inserted into the manuscript text.

Finally, the language still needs to be tidied up quite a bit, but I leave this to the editor to determine by how much. Apart from this the manuscript has been greatly improved by the revisions.

Reviewer #3 (Remarks to the Author):

This interesting manuscript by Zhou et al. investigates the effects of two stages artificial selection on the genome of ducks, one during their domestication and one during a more recent improvement of certain economically advantageous characteristics. Using population genomics analyses and an F2 intercross between the wild and improved breeds, the authors identify candidate divergence regions and proceed to map genetic variations affecting expression of two genes related to specific selected phenotypes.

Q1: For one of these genes (IGF2BP1), the selected region appears to be located around 150 kb away from the affected gene itself, thus the authors speculate that this might be due to impaired or altered long-range regulation of IGF2BP1. They then use Hi-C and collinearity analysis to assess the integrity of their genome assembly and check for evidence of large inversions that might create a false suggestion of long-range effects, without finding any.

Although the resolution of the Hi-C is quite limited, it is sufficient to validate the quality of the assembly and to support the lack of any large-scale inversions, as the authors appropriately state in the text ("...a high quality scaffold and no obvious inversion"). It should however be noted that this level of resolution cannot guarantee absence of smaller-scale inversions.

More importantly, it is unclear what breed they generated their Hi-C library from. The only information I could find in the methods was that "Duck livers were cross-linked...", but were these mallards? Pekin ducks? some of the F2 population? If the analysis was performed in only one of the parental breeds, then the other could still theoretically carry any kind of rearrangements, including an inversion in the IGF2BP1 region (although this is perhaps unlikely given the recombination events observed).

In general, there should be more stats about the Hi-C libraries in the methods section or as a supplementary figure/table, indicating for example the total number of reads sequenced, number of reads filtered out at every step, number of reads or read pairs that were used to build the contact maps after all filtering and intra-/inter-chromosomal interaction ratio as an estimator of library quality.

Q2: While it is not necessarily within the scope of the paper, it would have indeed been interesting to see whether the differences in the expression of IGF2BP1 were due to an alteration in the long-range interactions that this gene engages in. At present there is relatively little literature on the impact of 3D genome architecture on genome evolution and showing whether this level of regulation might have had an influence in this relatively well-defined setting of duck genomic evolution would have given an intriguing contribution to the understanding of evolutionary dynamics.

That being said, I understand that the authors tried to tackle this problem from their Hi-C data but

couldn't find any clear interactions between the IGF2BP1 promoter and the variant region (possibly due to the insufficient resolution) and I appreciate that the poor annotation of the genome can be a big hindrance in this kind of experiment.

Addressing this question would probably require the use of 4C using the IGF2BP1 promoter region as viewpoint in animals with opposing phenotypes. The structural information thus obtained could in turn allow for potential examination of the interacting sites for chromatin marks and for the binding of conserved architectural proteins (e.g. CTCF, cohesin) that can be influenced by simple modification of the sequence. This endeavour would require considerable additional sequencing and analysis and possibly even production of new duck-specific antibodies.

For these reasons and since the main focus of the paper is indeed different, I am not suggesting that the authors perform these experiments for this work. However, it may be an interesting avenue of research to explore in the future and it should be included as a point of discussion.

Q3 (minor): In the introduction, the white plumage of the Pekin ducks is characterised as "favourable", but the authors never really explain why plumage colour would constitute an economically positive trait. This is a rather accessory point, of course, but given that part of the findings presented concerns a gene controlling plumage colour, clarifying the role that this trait might have had in the selection process might help to better illustrate the logic behind the interpretation of the data.

Q4 (very minor): While I do appreciate some variety in the sometimes dull vocabulary of scientific literature, some of the language in the paper is a bit too hyperbolic. I could also find a typographic error.

- In the introduction I would change the "notable and superb" economic traits to "desirable" or "superior".

- In the MITF-B vs. MITF-M analysis section, in the parentheses: "careful *examination*" instead of "careful examination".

- At the end of the discussion, I am not sure what the authors mean when they say that "the creation of new traits occurred quickly and *inconceivably* in the comparatively short domestication process". If they simply want to stress (strongly) that the rate of emergence of these new traits is unexpectedly fast, I would maybe say "the creation of new traits occurred surprisingly quickly, as it was not believed to be possible in the comparatively short domestication process", although I cannot comment on the accuracy of such statement.

Reviewers' comments:

Reviewer #1 (Remarks to the Author):

The authors have very much improved the manuscript and addressed most of my comments. The paper now reads very well and I very much like the extensive results presented by the authors.

Response: Thank you for your positive comment. We appreciate your previous suggestions, which have helped us to greatly improve the manuscript.

Q1: Nevertheless, I still am not convinced by the conclusion from the authors that they have conclusive evidence that a mutation in a regulatory region affecting the *IGF2BP1* gene is the causative mutation for the phenotypic differences on growth identified between Peking duck and mallard.

On page 8, line 25 they refer to the Suppl. Table 14. However this table only shows the primers used for the qPCR but NOT the expression levels for the 10 different haplotypes. I could not find any information about the actual gene expression within the documents provided. Furthermore, the 100 kb region contains many other genes (HOX genes) and the authors do not show a direct correlation between the phenotype and the gene expression of *IGF2BP1*. In their cross, there is much LD and they cannot rule out that other genes underlie their observed phenotypic difference. In my view the authors need to (1) provide the gene expression data for the 10 haplotypes and (2) tone down their claims that a mutation in a regulatory sequence for the *IGF2BP1* is the causative mutation.

In the discussion (page 10, line 20) the authors refer this as “Another putative new mutation in the long-distance regulatory region”. Assuming that this region is indeed a long-distance regulator (of which I am not convinced), than they still do not identify the actual mutation within the 100 kb of sequence.

*Response: Thanks for your suggestion. We have provided the qPCR results for *IGF2BP1* in 10 recombinant types of 121 F₂ individuals (Supplementary Table 20, the source data of Fig. 4c) and *IGF2BP1* in 192 tissue samples of various types from mallards and Pekin ducks (Supplementary Table 19, the source data of Fig. 4a).*

*We have also toned down our claim and revised it as follows: “a **putative** long-distance regulatory mutation leading to continuous expression of the *IGF2BP1* gene after birth and an increase in body size by 15% and in feed efficiency by 6%” in the abstract and “Another putative new mutation **near** *IGF2BP1*, fixed under continual selection, acted as a major-effect QTL and transformed production performance” in the Discussion.*

The English language has now been very much improved, but I have identified a few minor mistakes which are listed below.

Page 2, line 5: “segregated for fine mapping”. To me this still doesn't sound correct

Page 3, line 6: “fowl” should be “fowls”

Page 3, last sentence: Z and W chromosomes (plural)

Page 4, line 4: replace “for a mean depth” by “at a mean depth”. I also think that the word “coverage” is redundant and should be removed.

Page 6, line 9: “association” instead of “associate”

Page 7, line 5: replace “was” by “is”

Page 7, line 23: change to: “the carcass traits measured in the”

Page 10, line 21-22: “which is fixedproductive performance”. This still is a weird sentence: “fixed under continual selection as a major effect QTL”? and “reinvented production performance”?

Page 11, line 15: replace “shared” by “for sharing”

Response: Thank you for patiently reviewing our manuscript and for noting the mistakes. In the revised manuscript, we have corrected each of these mistakes as suggested.

Reviewer #2 (Remarks to the Author):

I have now looked over the revisions and most have been met to my satisfaction. A few rather minor points are indicated below, however.

Q3. For the sentence that has been revised “the MITF-M isoform has almost no expression in ducks, strongly suggesting that the splicing changes in MITF were most likely caused by insertion and resulted in white plumage in Pekin ducks”, reword to the following:

“the MITF-M isoform has almost no expression in ducks, suggesting that the splicing changes in MITF were most likely caused by insertion and resulted in white plumage in Pekin ducks. However functional verification of this potentially causal mechanism is still required.”

Response: *Thank you for your suggestion. In the revised manuscript, we have added a sentence acknowledging the need for functional verification.*

Q5. I couldn't supplementary dataset 4 for the full GWAS results, but in any case, these should be given in a table in the supplementary section.

Response: *Thank you for your suggestion. The full GWAS results for 9 traits were approximately 2000 lines long, exceeding the reasonable range of length for a supplementary table; therefore, the top 10 significant loci have been included as Supplementary Table 15, and the loci that passed the significance threshold have been included as Supplementary Dataset 4.*

Q10. I don't think the authors properly understood my question. As I understand it, they classed sweeps as having to meet both F_{ST} and π thresholds. However, how many sweeps would be detected if they just used π (i.e. where a lot of sweeps discarded as they met the π requirements but not the F_{ST} thresholds)?

Response: *We apologize for misunderstanding your question. We performed class sweeps as having to meet both F_{ST} and π thresholds. The number of sweeps(40-kb windows) that were identified by π in the domestication and improvement stage, respectively, were 2513 and 2229(correspond to CDRs were 226 and 200), including 1288 and 1864 sweeps(correspond to 103 and 136 CDRs) that were identified by π but not by F_{ST} (see Supplementary Fig. 7).*

Q12. The authors show that the difference in FSTs between domestication and improvement stages is significant, but this p-value should be inserted into the manuscript text.

Response: Thank you for the reminder. We have altered the sentence as follows: “The values of F_{ST} were significantly higher in the CDRs of the improvement stage than in those of the domestication stage ($P = 3.25 \times 10^{-25}$).”

Finally, the language still needs to be tidied up quite a bit, but I leave this to the editor to determine by how much. Apart from this the manuscript has been greatly improved by the revisions.

Response: The manuscript has been carefully edited by a professional English editing service. We are trying our best to meet the language demands of the journal.

Reviewer #3 (Remarks to the Author):

This interesting manuscript by Zhou et al. investigates the effects of two stages artificial selection on the genome of ducks, one during their domestication and one during a more recent improvement of certain economically advantageous characteristics. Using population genomics analyses and an F2 intercross between the wild and improved breeds, the authors identify candidate divergence regions and proceed to map genetic variations affecting expression of two genes related to specific selected phenotypes.

Q1: For one of these genes (IGF2BP1), the selected region appears to be located around 150 kb away from the affected gene itself, thus the authors speculate that this might be due to impaired or altered long-range regulation of IGF2BP1. They then use Hi-C and collinearity analysis to assess the integrity of their genome assembly and check for evidence of large inversions that might create a false suggestion of long-range effects, without finding any.

Although the resolution of the Hi-C is quite limited, it is sufficient to validate the quality of the assembly and to support the lack of any large-scale inversions, as the authors appropriately state in the text ("...a high quality scaffold and no obvious inversion"). It should however be noted that this level of resolution cannot guarantee absence of smaller-scale inversions.

More importantly, it is unclear what breed they generated their Hi-C library from. The only information I could find in the methods was that "Duck livers were cross-linked...", but were these mallards? Pekin ducks? some of the F2 population? If the analysis was performed in only one of the parental breeds, then the other could still theoretically carry any kind of rearrangements, including an inversion in the IGF2BP1 region (although this is perhaps unlikely given the recombination events observed).

In general, there should be more stats about the Hi-C libraries in the methods section or as a supplementary figure/table, indicating for example the total number of reads sequenced, number of reads filtered out at every step, number of reads or read pairs that were used to build the contact maps after all filtering and intra-/inter-chromosomal interaction ratio as an estimator of library quality.

Response: *Thank you for your advice. We used a male Pekin duck to generate three independent Hi-C libraries. We have added this information to the Methods section (see "High-throughput chromosome conformation capture (Hi-C) experiment and sequencing" in the methods section). Considering the possible rearrangements, we have also toned down our claim and revised "a long-distance regulatory mutation" to "a **putative** long-distance regulatory mutation" in the abstract. Regarding the limited resolution of Hi-C and its use to validate the quality of the assembly, we have revised our claim as "we found a high-quality scaffold order assignment and no obvious **large-scale** inversions."*

In our supplementary materials, we have provided the statistical results of reads filtered out at every step (see Supplementary Table 21).

Supplementary Table 21. Summary of Hi-C data in duck. *This table shows the number of Hi-C sequencing reads in the category of each processing step in HiC-Pro.*

	Read count
Total read pairs	1,063,446,129
Uniquely aligned read pairs	554,436,799
Dumped read pairs	16,248,309
Self-circle	824,009
Dangling-end	155,542,662
Valid interactions	347,550,446
Filtered valid interactions	129,168,069
Intrachromosomal contacts	71,680,763
Interchromosomal contacts	57,487,306

Q2: While it is not necessarily within the scope of the paper, it would have indeed been interesting to see whether the differences in the expression of IGF2BP1 were due to an alteration in the long-range interactions that this gene engages in. At present there is relatively little literature on the impact of 3D genome architecture on genome evolution and showing whether this level of regulation might have had an influence in this relatively well-defined setting of duck genomic evolution would have given an intriguing contribution to the understanding of evolutionary dynamics.

That being said, I understand that the authors tried to tackle this problem from their Hi-C data but couldn't find any clear interactions between the IGF2BP1 promoter and the variant region (possibly due to the insufficient resolution) and I appreciate that the poor annotation of the genome can be a big hindrance in this kind of experiment.

Addressing this question would probably require the use of 4C using the IGF2BP1 promoter region as viewpoint in animals with opposing phenotypes. The structural information thus obtained could in turn allow for potential examination of the interacting sites for chromatin marks and for the binding of conserved architectural proteins (e.g. CTCF, cohesin) that can be influenced by simple modification of the sequence. This endeavour would require considerable additional sequencing and analysis and possibly even production of new duck-specific antibodies.

For these reasons and since the main focus of the paper is indeed different, I am not suggesting that the authors perform these experiments for this work. However, it may be an interesting avenue of research to explore in the future and it should be included as a point of discussion.

Response: *Thank you for your insightful comments and suggestions. As suggested, we will continue working to identify the causative mutation that leads to continuous expression of the*

IGF2BP1 gene after hatching, leading to a 15% increase in body size and a 6% increase in feed efficiency.

Q3 (minor): In the introduction, the white plumage of the Pekin ducks is characterised as "favourable", but the authors never really explain why plumage colour would constitute an economically positive trait. This is a rather accessory point, of course, but given that part of the findings presented concerns a gene controlling plumage colour, clarifying the role that this trait might have had in the selection process might help to better illustrate the logic behind the interpretation of the data.

Response: *Thank you for noting this ambiguous description. There were two main reasons why we characterized white plumage as an "economic trait". First, the white feathers favored by humans are generally used to make down jackets or quilts, and they are also more suitable for dyeing than black or tan feathers. Second, a lack of pigment increases the appeal of a carcass for culinary use. We revised the sentence as follows: "Pekin ducks show many striking changes such as white plumage, a favorable feature that meets the demand for white down as a filler for jackets or quilts and makes the carcass easy to clean".*

Q4 (very minor): While I do appreciate some variety in the sometimes dull vocabulary of scientific literature, some of the language in the paper is a bit too hyperbolic. I could also find a typographic error.

- In the introduction I would change the "notable and superb" economic traits to "desirable" or "superior".
- In the MITF-B vs. MITF-M analysis section, in the parentheses: "careful *examination*" instead of "careful examination".
- At the end of the discussion, I am not sure what the authors mean when they say that "the creation of new traits occurred quickly and *inconceivably* in the comparatively short domestication process". If they simply want to stress (strongly) that the rate of emergence of these new traits is unexpectedly fast, I would maybe say "the creation of new traits occurred surprisingly quickly, as it was not believed to be possible in the comparatively short domestication process", although I cannot comment on the accuracy of such statement.

Response: *Thank you for your detailed revision comments, and please excuse our occasional poor descriptions and spelling errors. Following your suggestions, we have corrected these mistakes in the revised manuscript.*

REVIEWERS' COMMENTS:

Reviewer #1 (Remarks to the Author):

With the expression data now provided in supplementary tables 19 and 20 and by somewhat toning down the conclusion by referring to the identified mutation as "putative", the authors have sufficiently addressed my remaining concern I raised in my previous review.

I think that the English of the manuscript can still be improved, but I leave that to the discretion of the editors of the journal.

Reviewer #2 (Remarks to the Author):

The authors have now fully addressed all of my concerns. I also have no problems in signing my review. Well done on an excellent study
sincerely

Dominic Wright

Reviewer #3 (Remarks to the Author):

In their review, the authors have addressed all my previous concerns to my satisfaction.

For clarity purposes, however, I would suggest that they adjust the text at page 9 to: "Considering that errors in scaffold genome assembly such as inversions could cause an illusion of long-distance regulation, we examined the end of chromosome 28 using high-throughput chromosome conformation capture (Hi-C) data and collinearity analyses (Supplementary Fig. 15 and Supplementary Table 21); we found a high quality scaffold order assignment with no evidence of obvious large-scale inversions". It would also be advisable to specify that both the genome assembly and the Hi-C libraries were prepared from the same subspecies (Pekin ducks).

This is just to avoid giving the impression that the Hi-C data is indicating the absence of inversions between mallards and Pekin ducks and state more unambiguously that this data is being used here exclusively to validate the genome assembly.

Aside from this, I found this paper very interesting and I look forward to potentially reading their follow-up characterising the role of three-dimensional architecture on the evolution of the duck genome.

Reviewer #1 (Remarks to the Author):

With the expression data now provided in supplementary tables 19 and 20 and by somewhat toning down the conclusion by referring to the identified mutation as "putative", the authors have sufficiently addressed my remaining concern I raised in my previous review.

I think that the English of the manuscript can still be improved, but I leave that to the discretion of the editors of the journal.

Response: Thank you for your positive comment. We appreciate your previous suggestions, which have helped us to greatly improve the manuscript.

Reviewer #2 (Remarks to the Author):

The authors have now fully addressed all of my concerns. I also have no problems in signing my review. Well done on an excellent study
sincerely
Dominic Wright

Response: Thank you for patiently reviewing our manuscript.

Reviewer #3 (Remarks to the Author):

In their review, the authors have addressed all my previous concerns to my satisfaction.

For clarity purposes, however, I would suggest that they adjust the text at page 9 to: "Considering that errors in scaffold genome assembly such as inversions could cause an illusion of long-distance regulation, we examined the end of chromosome 28 using high-throughput chromosome conformation capture (Hi-C) data and collinearity analyses (Supplementary Fig. 15 and Supplementary Table 21); we found a high quality scaffold order assignment with no evidence of obvious large-scale inversions".

It would also be advisable to specify that both the genome assembly and the Hi-C libraries were prepared from the same subspecies (Pekin ducks).

This is just to avoid giving the impression that the Hi-C data is indicating the absence of inversions between mallards and Pekin ducks and state more unambiguously that this data is being used here exclusively to validate the genome assembly.

Response: Thank you for your suggestion. In the revised manuscript, we have replaced this sentence as you revised..

Aside from this, I found this paper very interesting and I look forward to potentially reading their follow-up characterising the role of three-dimensional architecture on the evolution of the duck genome.

Response: Thank you for your insightful comments and suggestions.